# Morphometric based differentiation among *Trichogramma* spp.

**Salman Khan**[ID]¤*, **Mohd. Yousuf, Mohsin Ikram**[ID]

Forest Entomology Discipline, Forest Protection Division, Forest Research Institute, Dehradun, Uttarakhand, India

¤ Current address: Forest Survey of India, Central Zone, Seminary Hills, Nagpur, Maharashtra, India
* salman1315@gmail.com

**Data Availability Statement:** All relevant data are within the manuscript.

**Funding:** The author(s) received no specific funding for this work.

## Abstract

Genus *Trichogramma* Westwood is being utilized in various biological control programme worldwide. *Trichogramma* spp. are egg parasitoids which lay their eggs inside the host eggs. More than 240 species are known, of which, 45 species are recorded from India. It is quite difficult to identify and differentiate among the species of *Trichogramma* due to smaller size. In present study, we hypothesized a methodology to identify the *Trichogramma* species based on potential morphometric characteristics. The males and females of five different species of *Trichogramma* namely, *T. brassicae*, *T. chilotraeae*, *T. danaidiphaga*, *T. danausicida* and *T. dendrolimi* were morphometrically analysed. A total of 33 characters in males and 27 in females were taken into consideration for morphometric identification and analysis using principal component method. It was observed that few characters of male like body length, length of flagellar hair, maximum marginal hair length of fore wing, genitalia characters including aedeagus length, distance between chelate structure and gonoforceps, and others were identified as important morphometric characters. Similarly, in females, ovipositor length, body and head length, eye width, antennal club width and number of setae in forewings were found important for taxonomic identification. Male and female of each species are clearly identified from high definition microscopic images.

## Introduction

Insects play an important role in the biological system including pollination, regulation of population density of pests and many others [1] but at the same time, some insects have negative impact on plants and food chains. So many insect-pests are observed on forest trees, of which leaf defoliators, stem borers, sap-sucker, termites, crickets, hoppers cause huge economic loss. The damage caused by these insect-pests is quite difficult to control and manage in trees due to more tree height. However, some chemicals are used in nursery and sapling stage of plant. The use of such a mechanism for insect control is creating an adverse impact on climate and human health. So, to ensure that minimum effect to be levied on nature and human health, we have to adopt either silvicultural and mechanical operations or biological agents to control Insect-pests [2]. Genus *Trichogramma* is renowned for its wide use in the biological controls

**Competing interests:** The authors have declared that no competing interests exist.

of important insect pests of agriculture and forestry importance worldwide. *Trichogramma* spp. are biological control agents and classified as egg parasitoids [3] as they lay their egg into the host eggs [4]. It is an established fact that these parasitoids were exploited for controlling the several insect pests, belonging to the various orders such as Coleoptera, Hymenoptera and Lepidoptera [5, 6, 7, 8]. Genus *Trichogramma* Westwood belongs to the superfamily Chalcidoidea and family Trichogrammatidae [9–11], which is characterized by having three tarsal segments. *Trichogramma* spp. are being characterized by having female antennae with 2-segmented funicle and one segmented club; discal setae arranged in rows; forewings with sigmoid venation and presence of RS1 vein track [6, 12, 13]. The differentiation among the species genus *Trichogramma* is quite difficult owing to their smaller size (<0.5mm) and low interspecific morphological characters diversity creating many identification problems [14].

Morphometric is basically, the qualitative and quantitative analysis of the form of any organism [15], or analysis of morphological variations [16] and also known as numerical taxonomy [17]. As the species of *Trichogramma* are very minute, even they cannot be seen easily with naked eyes. To distinguish among the various species of *Trichogramma*, various researchers have approached to formulate the keys but still, confusion exist [18]. In males of *Trichogramma*, the identification is mostly based on genitalia characteristics, antennae and forewings [19, 20, 21] whereas, in females, ovipositor and hind tibia length are considered for identifying different species. Various researchers such as Pinto and Oatman [22], Nagaraja [23], Nagaraja and Gupta [24], Burks and Heraty [25], Yousuf *et al.* [26], Nagaraja and Mohanraj [27, 28] & Fursov and Kodan [29] have provided taxonomic approaches for identification of *Trichogramma* spp. On the basis of morphometric characters of genitalia, wings, and hind leg. Here again, various important characters were not taken into consideration [29]. In present study, we studied the male and female of five different *Trichogramma* species. It is to propose the morphometric and traditional basis for identification and differentiation among *Trichogramma* spp. which are very important in forest tree species.

## Material and method

### Culture of *Trichogramma* spp.

Five *Trichogramma* spp. namely, *T. brassicae* Bezdenko, *T. chilotraeae* Nagaraja and Nagarkatti, *T. danaidiphaga* Nagaraja and Prashanth, *T. danausicida* Nagaraja and *T. dendrolimi* Matsumura were studied for morphometric analysis due to the following reasons: 1) large host range for *T. brassicae* and *T. dendrolimi*, 2) availability of *T. chilotraeae*, *T. danaidiphaga* and *T. danausicida* from India, and 3) cultures availability of both sexes of these species from India for rearing and morphometric analysis. Culture of these five *Trichogramma* spp. were procured from the National Bureau of Agricultural Insect Resources (NBAIR), Bangalore, India. The cultures of *Trichogramma* were maintained on the eggs of *Corcyra cephalonica* under control conditions i.e. temperature @27±1˚C and relative humidity @65% by following the methodology suggested by Jalali *et al.* [30] and Nathan *et al.* [31]. The cultures were also submitted to the National Forest Insect Collection (NFIC), Forest Research Institute Dehradun, India for authentic identification and cross verification of the species. The cultures of *Trichogramma* spp. were further confirmed by using the literatures available [21, 25, 32–34].

### Dissection and slide preparations

The specimens of *Trichogramma* were stored in 70% Ethanol followed by soaking into Potassium Hydroxide (KOH) 10% for 5 minutes and in 10% Glacial Acetic Acid for 5 minutes. Then, the specimens were treated with distilled water followed by Ethanol treatment in a dehydration series of 30%, 50%, 70%, 90% and absolute alcohol for 5 minutes at each step [35].

After treatment with Ethanol, the specimens were kept in Clove oil for 2 minutes. The dissections of *Trichogramma* specimens were carried out in the Clove oil medium using the stereoscopic zoom binocular microscope (AARK-Zoom Star-VI). The slides of each specimens were prepared using the methodology given by Platner *et al.* [36], Yousuf *et al.* [26] & Hassan and Yousuf [37]. A total 50 males and 50 females of each species of *Trichogramma* were dissected to check the morphometric variability among the species. Dissections of males *Trichogramma* specimens were carried out for Head, Forewings, Hindwings, Hind tibia and Genitalia. Similarly, the females of *Trichogramma* specimens were dissected for Head, Forewings, Hindwings, Hind tibia and Ovipositor. Also, male and female antennae were dissected out for flagellum and club studies. Further, treating with the normal series of dehydration followed by clearing with Clove oil for one hour and then mounted in Canada Balsam/Euparol on slides [37, 38]. Then the slides were kept overnight @ temperature 27±1˚C and 65% R.H.

## Morphometric observation

The terminology used is as per the earlier research work of Yousuf and Shafee [6], Hassan and Yousuf [37] & Khan *et al.* [39]. The details of the morphometric characters taken into consideration were given in Fig 1 and Table 1.

## Photography

After the measurements of different characters of *Trichogramma* spp., photography was carried out, using Nikon Digital Sight DS-Fi1 attached with Nikon Optiphot microscope.

## Statistical analysis

The morphometric observations were statistically analysed using SPSS software, Microsoft Excel, Minitab-18 and XLSTAT. Principal Component Analysis (PCA) was employed for the morphometric analysis.

## Results and discussion

The results are based on the morphometric features collected for the males and females of *Trichogramma brassicae*, *T. chilotraeae*, *T. danaidiphaga*, *T. danausicida* and *T. dendrolimi*.

### *Trichogramma brassicae* Bezdenko, 1968: 36

**Diagnosis.** Males are yellowish in colour and smaller than female. Flagellar hairs almost half of the flagellar length with 25–33 hairs. Fore wings length is 2 times of its width. Dorsal lamina with deep notch, end of MVP (Median Ventral Projection) not reaching that of dorsal lamina. Dorsal blade of genitalia strongly incised, the distance separating the apex of the blade from the level of its greater width equal to this width. The results are in accordance with the observation recorded by Bohinc *et al.* [40], Chiriac [41] & Viggiani and Laudonia [42].

### Morphometric measurements of *T. brassicae*

The BL in *T. brassicae* varied from 0.4464 mm to 0.5022 mm in males and 0.4743 mm to 0.5208 mm in females (Table 2). The HTL and HTW in the males of *T. brassicae* ranged from 0.1576 mm to 0.1686 mm and 0.0220 mm to 0.0293 mm respectively. Earlier, Scholler and Agamy [43] reported the similar results for the tibial characteristics. The average GCL, GCW, CTG and AL measured as 0.1350±0.007 mm, 0.0499±0.003 mm, 0.0168±0.001 mm and 0.1313 ±0.007 mm, respectively. The similar genitalia characteristics were reported by Chiriac [41]. The mean OL in the females of *T. brassicae* was 0.1727±0.007 mm. Kostadinov and Pintureau

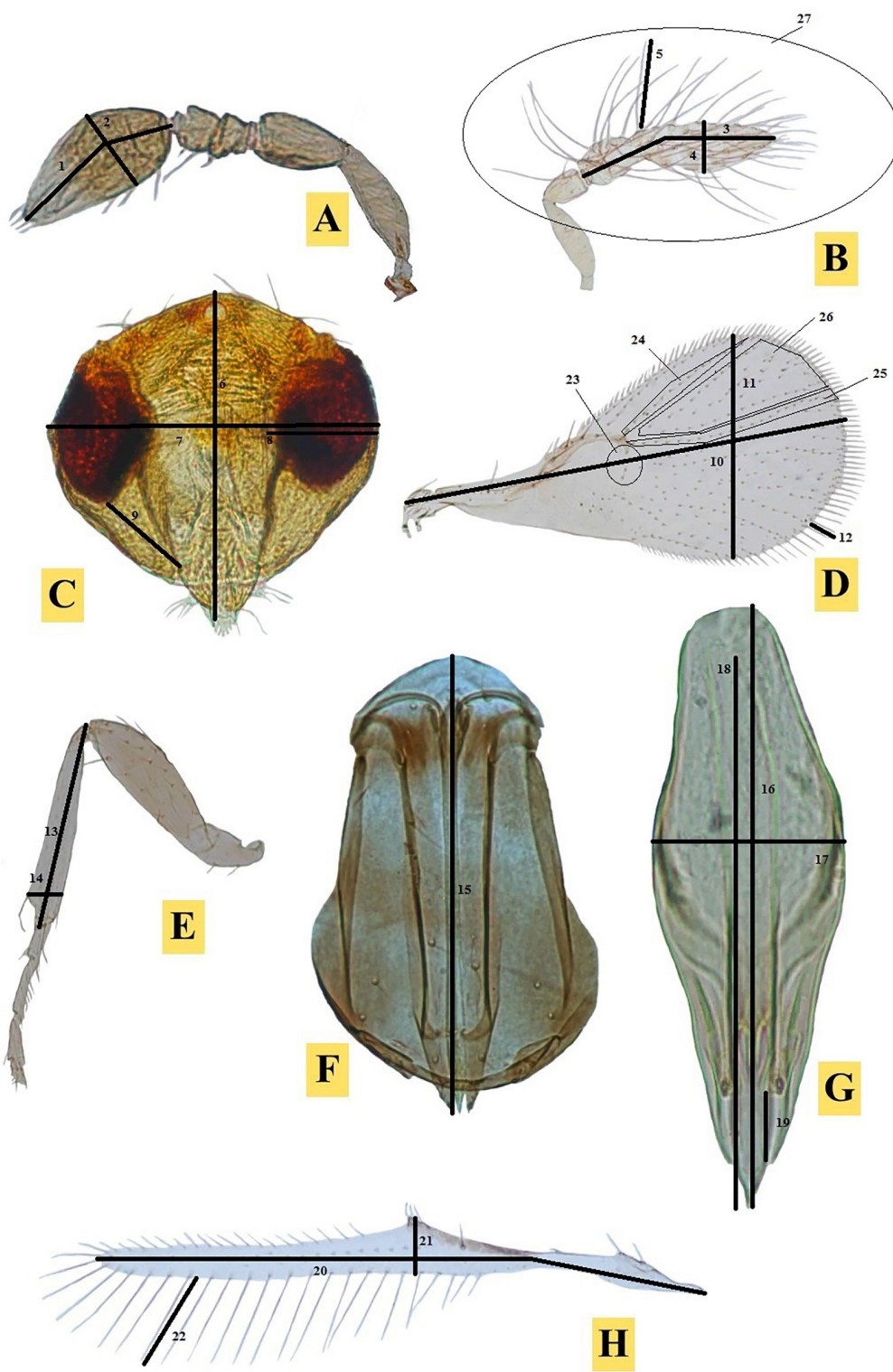

**Fig 1. Morphometric characters being used in the morphometric analysis of *Trichogramma* spp.** (A-H): A-Antennal club ♀; B-Antennal flagellum ♂; C-Head; D-Forewing; E-Hind tibia; F-Female ovipositor; G-Male genitalia; and H-Hindwing.

**Table 1. Characters of male/females studied for morphometrics of *Trichogramma* spp.**

| S. No. | Abbreviations | Male/Female characters | Line number on Fig 1 |
|---|---|---|---|
| 1 | ACL | Antennal Club length | 1 |
| 2 | ACW | Antennal Club width | 2 |
| 3 | FL | Flagellum length | 3 |
| 4 | FW | Flagellum width | 4 |
| 5 | FHL | Flagellar hair length (longest) | 5 |
| 6 | HL | Head length | 6 |
| 7 | HW | Head width | 7 |
| 8 | EW | Eye width | 8 |
| 9 | MS | Malar space | 9 |
| 10 | FWL | Fore wings length | 10 |
| 11 | FWW | Fore wings width | 11 |
| 12 | MFW | Maximum length of marginal fringe of Fore wings | 12 |
| 13 | HTL | Length of hind tibia | 13 |
| 14 | HTW | Width of hind tibia | 14 |
| 15 | OL | Ovipositor length | 15 |
| 16 | GCL | Genital capsule length | 16 |
| 17 | GCW | Genital capsule width | 17 |
| 18 | AL | Aedeagus length | 18 |
| 19 | CTG | Distance from CS to GF | 19 |
| 20 | HWL | Hind wings length | 20 |
| 21 | HWW | Hind wings width | 21 |
| 22 | MHW | Maximum length of marginal fringe of Hind wings | 22 |
| 23 | RS1 | Fore wings setae in RS1 vein track | 23 |
| 24 | RS2 | Fore wings setae in RS2 vein track | 24 |
| 25 | RM | Fore wings setae in r-m vein track | 25 |
| 26 | RR | Fore wings setae in RS2 & r-m vein track | 26 |
| 27 | NFH | No. of flagellar hairs | 27 |
| 28 | FL/HTL | Flagellar length/Hind tibia length | 3/13 |
| 29 | FL/FW | Flagellar length /Flagellar width | ¾ |
| 30 | FHL/FW | Flagellar hairs length (longest)/Flagellar width | 15/4 |
| 31 | FWL/FWW | Fore wings length/Fore wings width | 10/11 |
| 32 | FWW/MFW | Fore wings width/Longest marginal fringe of fore wings | 11/12 |
| 33 | GCL/HTL | Genital capsule length/Hind tibia length | 16/13 |
| 34 | GCL/GCW | Genital capsule length/Genital capsule width | 16/17 |
| 35 | OL/ACL | Ovipositor length/Antennal club length | 15/1 |
| 36 | ACL/ACW | Antennal club length/Antennal club width | 1/2 |
| 37 | OL/HTL | Ovipositor length/Hind tibia length | 15/14 |
| 38 | HWW/MHW | Hind wings width/Longest marginal fringe of hind wings | 21/22 |

[44] discussed the morphological characteristics of *T. brassicae* and compared with *T. evanescens* and reported similar kind of results. The ratio of FHL and FW in males of *T. brassicae* was estimated from 2.4000 to 3.7143 (Table 3). The FHL was about as long as 3x of FW which is similar in the range as reported by Chiriac [41]. Del Pino *et al.* [45] discussed the FHL/FW as 3.6–4.1, which is similar with our result. Pintureau and Voegele [46] also reported the similar structure of flagellum as recorded in present study. Images for dissected characters are presented in Figs 2 and 3.

**Table 2. Morphometrics of main characters for males and females of *Trichogramma brassicae* Bezdenko.**

| S. No. | Characters | *T. brassicae* (Male) | *T. brassicae* (Female) |
|---|---|---|---|
| 1 | BL | 0.4864±0.016 | 0.4994±0.015 |
|  |  | (0.4464–0.5022) | (0.4743–0.5208) |
| 2 | HL | 0.1651±0.010 | 0.1748±0.012 |
|  |  | (0.1495–0.1817) | (0.1587–0.1955) |
| 3 | HW | 0.1842±0.010 | 0.1980±0.008 |
|  |  | (0.1702–0.2001) | (0.1863–0.2093) |
| 4 | EW | 0.0828±0.003 | 0.0881±0.003 |
|  |  | (0.0782–0.0874) | (0.0828–0.0943) |
| 5 | MS | 0.0515±0.002 | 0.0501±0.002 |
|  |  | (0.0483–0.0552) | (0.0483–0.0552) |
| 6 | FL/ACL | 0.1599±0.007 | 0.0829±0.006 |
|  |  | (0.1503–0.1760) | (0.0733–0.0880) |
| 7 | FW/ ACW | 0.0308±0.003 | 0.0330±0.004 |
|  |  | (0.0256–0.0367) | (0.0293–0.0403) |
| 8 | FHL | 0.0931±0.004 | - - - - - - - - - - - - |
|  |  | (0.0843–0.0990) |  |
| 9 | FWL | 0.5143±0.016 | 0.5078±0.020 |
|  |  | (0.4929–0.5394) | (0.4836–0.5487) |
| 10 | FWW | 0.2474±0.008 | 0.2344±0.016 |
|  |  | (0.2325–0.2604) | (0.2046–0.2604) |
| 11 | MFW | 0.0343±0.002 | 0.0366±0.002 |
|  |  | (0.0322–0.0368) | (0.0322–0.0391) |
| 12 | HWL | 0.3804±0.005 | 0.3899±0.012 |
|  |  | (0.3749–0.3933) | (0.3726–0.4163) |
| 13 | HWW | 0.0377±0.002 | 0.0373±0.001 |
|  |  | (0.0345–0.0414) | (0.0368–0.0391) |
| 14 | MHW | 0.0580±0.002 | 0.0600±0.003 |
|  |  | (0.0552–0.0621) | (0.0552–0.0644) |
| 15 | HTL | 0.1632±0.005 | 0.1654±0.006 |
|  |  | (0.1576–0.1686) | (0.1577–0.1760) |
| 16 | HTW | 0.0246±0.002 | 0.0260±0.002 |
|  |  | (0.0220–0.0293) | (0.0220–0.0290) |
| 17 | GCL | 0.1350±0.007 | - - - - - - - - - - - - |
|  |  | (0.1288–0.1472) |  |
| 18 | GCW | 0.0499±0.003 | - - - - - - - - - - - - |
|  |  | (0.0414–0.0529) |  |
| 19 | CTG | 0.0168±0.001 | - - - - - - - - - - - - |
|  |  | (0.0161–0.0184) |  |
| 20 | AL | 0.1313±0.007 | - - - - - - - - - - - - |
|  |  | (0.1219–0.1449) |  |
| 21 | OL | - - - - - - - - - - - - | 0.1727±0.007 |
|  |  |  | (0.1633–0.1863) |
| 22 | RS1 | 2–4 | 2–4 |
| 23 | RS2 | 7–10 | 7–10 |
| 24 | RM | 16–21 | 18–24 |
| 25 | RR | 38–45 | 36–47 |

(*Continued*)

**Table 2.** (Continued)

| S. No. | Characters | *T. brassicae* (Male) | *T. brassicae* (Female) |
|---|---|---|---|
| 26 | NFH | 25–33 | - - - - - - - - - - - - |

Mean±standard deviation (top value) and range (parentheses); S. No. 1–21 are in mm and 22–26 are in numbers.

### *Trichogramma chilotraeae* Nagaraja and Nagarkatti, 1969: 394

**Diagnosis.** In males, the body is yellow in colour with blackish abdominal terga and hind coxae. Antennal flagellum with blunt hairs and three times longer with the width of flagellum. In genitalia, DEG (Dorsal Expansion of Gonobase) is triangular; not reaching the tips of GF but extend only to level of CS; CS located far below the level of tip of GF (Gonoforceps); MVP (Median Ventral Projection) is very distinct and long with two chitinous ridge (CR) extending anteriorly only for a short distance from MVP. Aedeagus and apodemes together shorter than the hind tibia. In females, anterior portion of mesoscutum is blackish in colour. In females, ovipositor slightly longer than the hind tibial length [6, 34, 47–50].

**Table 3. Morphometrics of ratios characters for males and females of *Trichogramma brassicae* Bezdenko.**

| S. No. | Characters | *T. brassicae* (Male) | *T. brassicae* (Female) |
|---|---|---|---|
| 1 | FHL/FW | 3.0587±0.400 | - - - - - - - - - - - - |
|   |   | (2.4000–3.7143) |   |
| 2 | FL/FW | 5.2429±0.622 | - - - - - - - - - - - - |
|   |   | (4.3000–6.1429) |   |
| 3 | FL/HTL | 0.9803±0.045 | - - - - - - - - - - - - |
|   |   | (0.8913–1.0435) |   |
| 4 | ACL/ACW | - - - - - - - - - - - - | 2.5628±0.460 |
|   |   |   | (1.9090–3.0000) |
| 5 | OL/ACL | - - - - - - - - - - - - | 2.0959±0.193 |
|   |   |   | (1.8555–2.3834) |
| 6 | HTL/ACL | - - - - - - - - - - - - | 2.0043±0.163 |
|   |   |   | (1.7917–2.2500) |
| 7 | FWL/FWW | 2.0791±0.022 | 2.1712±0.085 |
|   |   | (2.0385–2.1200) | (2.0769–2.3636) |
| 8 | FWW/MFW | 7.2332±0.393 | 6.4222±0.4700 |
|   |   | (6.3179–7.7981) | (5.8125–7.2205) |
| 9 | HWW/MHW | 0.6523±0.052 | 0.6217±0.031 |
|   |   | (0.5555–0.7200) | (0.5714–0.6667) |
| 10 | GCL/GCW | 2.7128±0.169 | - - - - - - - - - - - - |
|   |   | (2.5217–3.1111) |   |
| 11 | GCL/HTL | 0.8280±0.047 | - - - - - - - - - - - - |
|   |   | (0.7635–0.9043) |   |
| 12 | OL/HTL | - - - - - - - - - - - - | 1.0446±0.022 |
|   |   |   | (1.0090–1.0794) |

Mean±standard deviation (top value) and range (parentheses).

*Trichogramma brassicae* ♂

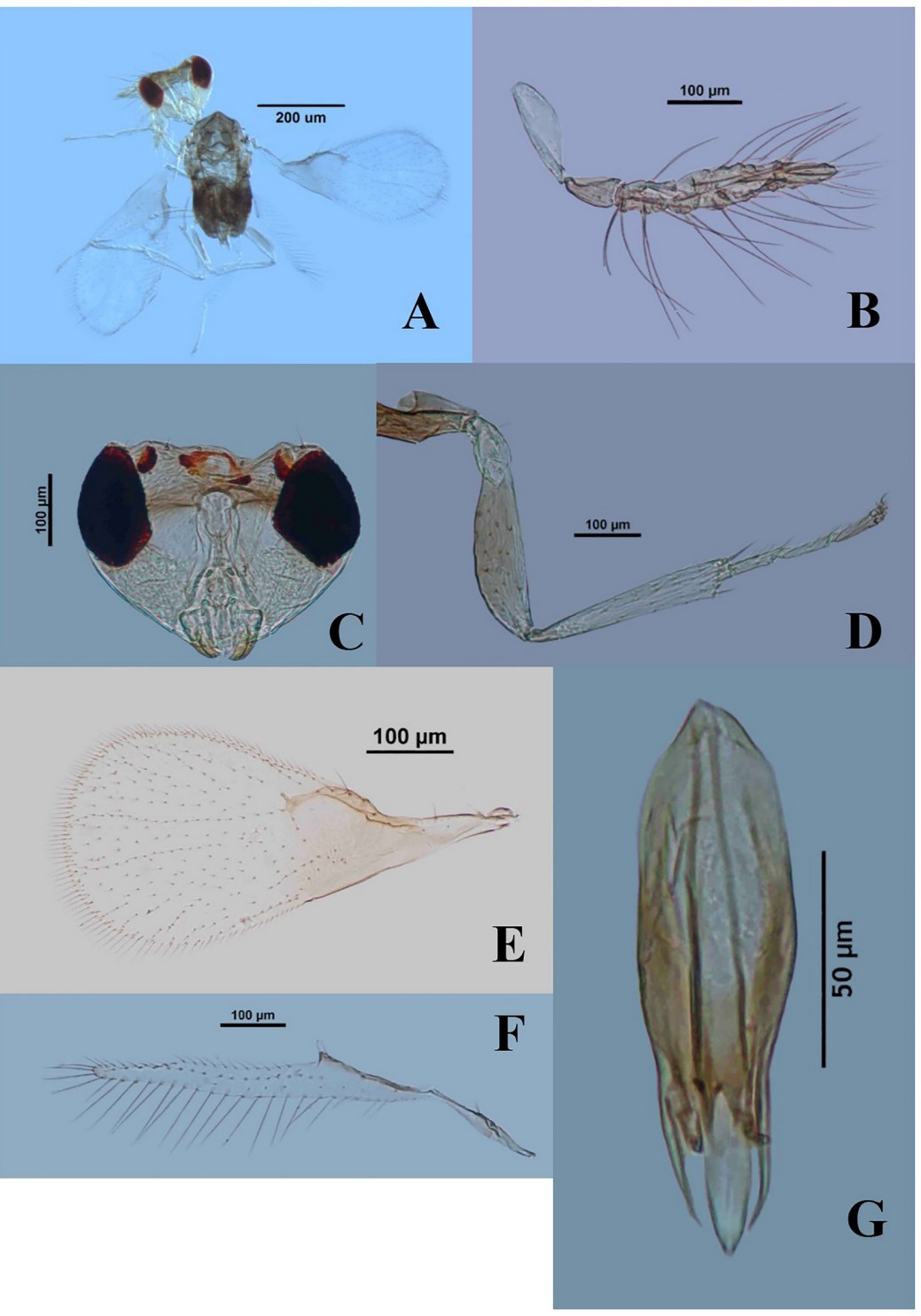

**Fig 2. Morphological characters of male *Trichogramma brassicae* Bezdenko.** A—whole body, B—antenna, C—head, D—hind tibia, E—fore wing, F—hind wing, G—genital capsule.

## Trichogramma brassicae ♀

**Fig 3. Morphological characters of female *Trichogramma brassicae* Bezdenko.** A—whole body, B—antenna, C—head, D—hind tibia, E—fore wing, F—hind wing, G—ovipositor.

## Morphometric measurements of *T. chilotraeae*

The males of *T. chilotraeae* were recorded smaller than the females (Table 4). The HL, HW, EW and MS in both female and males of *T. chilotraeae* were found almost equal. The HL and HW described by Nagaraja and Nagarkatti [34] are in accordance with the finding of present study. In the males, GCL, GCW, CTG and AL were measured as 0.1424±0.006 mm, 0.0478 ±0.002 mm, 0.0136±0.001 mm and 0.1371±0.006 mm, respectively. In females, the OL was measured as 0.1764±0.008 mm. Images for dissected characters are presented in Figs 4 and 5. The genitalia characteristics along with its structure for *T. chilotraeae* were also recorded by Ruiz and Korytkowski [49] and found similar kind of results. The ratio of FHL and FW varied from 2 to 3 (Table 5). In males, the FL was about 4.8167±0.743 times of FW. In females, the ACL was about 2.3349±0.176 times the ACW. The morphometric characteristics of flagellum in *T. chilotraeae* explained by Ruiz and Korytkowski [49] was found similar to the results obtained in the present study.

## *Trichogramma danaidiphaga* Nagaraja and Prashanth, 2010a: 297

**Diagnosis.** Antennae with scape broad at base and about 2.53x the pedicel; flagellum with ring segment, having 21–39 long hairs with blunt ends, longest about more than 2x of the maximum width of flagellum. Genitalia with DEG (Dorsal Expansion of Gonobase) having broad base, with narrow anterior extremity on both sides. MVP (Median Ventral Projection) short and rigid; CS (Chelate Structure) below the level of GF (Gonoforceps); Aedeagus with short apodemes, both together as long as total genital length and slightly shorter than hind tibia. Females antenna with pedicel slightly longer than funicle; club almost as long as the scape. Ovipositor comparatively of hind tibial length [27, 50].

## Morphometric measurements of *T. danaidiphaga*

The morphometric results of *T. danaidiphaga* depicted that the males are relatively smaller than the females. The HL, MS and HW were almost equal in both males and females of *T. danaidiphaga* (Table 6). In males, the mean GCL, GCW, CTG and AL were 0.1267±0.004 mm, 0.0389±0.001 mm, 0.0074±0.001 and 0.1244±0.004 mm, respectively. Yousuf *et al.* [50] have also discussed the structure of genitalia in *T. danaidiphaga* and this result found in compliance with them. In males, the ratio FHL/FW ranged from 2.1 to 2.6 (Table 7). Also, the FL was about 4.275±0.303 times of FW. Similarly, the FL was about 0.9117±0.054 times of HTL. According to the Nagaraja and Mohanraj [27], FHL/FW was 2.2, which is quite similar to the present results. Images for dissected characters are given in Figs 6 and 7.

## *Trichogramma danausicida* Nagaraja, 1996: 3

**Diagnosis.** In Males, the head is orange-yellow and prothorax and mesoscutellum are lightly infuscate. Flagellum is more than 1.8 times the length of scape, having long hairs, 2.2 times of the flagellum width. The genitalia, with narrow DEG (Dorsal Expansion of Gonobase) and pointed to the level of CS (Chelate Structure); also, CS is below the level of GF (Gonoforceps); CR (Central Ridge) is paired; Aedeagus is about the length of apodemes; both together slightly longer than hind tibial length. In females, the body is orange yellowish with dark fuscous on its abdominal terga; Antenna with club as 0.8 times the scape length; Ovipositor is longer than hind tibia [23, 27, 50].

**Table 4. Morphometrics of main characters for males and females of *Trichogramma chilotraeae* Nagaraja and Nagarkatti.**

| S. No. | Characters | *T. chilotraeae* (Male) | *T. chilotraeae* (Female) |
|--------|-----------|------------------------|--------------------------|
| 1 | BL | 0.4724±0.017 | 0.5143±0.015 |
|   |    | (0.4371–0.4929) | (0.4836–0.5394) |
| 2 | HL | 0.1985±0.012 | 0.1879±0.010 |
|   |    | (0.1863–0.2254) | (0.1725–0.2047) |
| 3 | HW | 0.2162±0.013 | 0.1990±0.010 |
|   |    | (0.1978–0.2415) | (0.1886–0.2185) |
| 4 | EW | 0.0925±0.004 | 0.0895±0.004 |
|   |    | (0.0874–0.0966) | (0.0851–0.0966) |
| 5 | MS | 0.0529±0.004 | 0.0541±0.003 |
|   |    | (0.0460–0.0598) | (0.0506–0.0598) |
| 6 | FL/ACL | 0.1613±0.007 | 0.0792±0.003 |
|   |    | (0.1540–0.1723) | (0.0733–0.0843) |
| 7 | FW/ ACW | 0.0341±0.005 | 0.0341±0.003 |
|   |    | (0.0257–0.0403) | (0.0293–0.0403) |
| 8 | FHL | 0.0781±0.002 | - - - - - - - - - - - - |
|   |    | (0.0733–0.0807) |  |
| 9 | FWL | 0.4901±0.015 | 0.4669±0.020 |
|   |    | (0.4650–0.5115) | (0.4464–0.5022) |
| 10 | FWW | 0.2353±0.009 | 0.2288±0.010 |
|   |    | (0.2232–0.2511) | (0.2139–0.2511) |
| 11 | MFW | 0.0361±0.002 | 0.0357±0.002 |
|   |    | (0.0345–0.0391) | (0.0322–0.0391) |
| 12 | HWL | 0.4060±0.015 | 0.3977±0.011 |
|   |    | (0.3772–0.4186) | (0.3795–0.4186) |
| 13 | HWW | 0.0380±0.002 | 0.0373±0.002 |
|   |    | (0.0368–0.0414) | (0.0345–0.0391) |
| 14 | MHW | 0.0598±0.002 | 0.0552±0.003 |
|   |    | (0.0552–0.0621) | (0.0506–0.0598) |
| 15 | HTL | 0.1654±0.006 | 0.1632±0.005 |
|   |    | (0.1577–0.1760) | (0.1540–0.1687) |
| 16 | HTW | 0.0242±0.002 | 0.0242±0.002 |
|   |    | (0.0220–0.0257) | (0.0220–0.0257) |
| 17 | GCL | 0.1424±0.006 | - - - - - - - - - - - - |
|   |    | (0.1334–0.1518) |  |
| 18 | GCW | 0.0478±0.002 | - - - - - - - - - - - - |
|   |    | (0.0460–0.0506) |  |
| 19 | CTG | 0.0136±0.001 | - - - - - - - - - - - - |
|   |    | (0.0115–0.0138) |  |
| 20 | AL | 0.1371±0.006 | - - - - - - - - - - - - |
|   |    | (0.1288–0.1472) |  |
| 21 | OL | - - - - - - - - - - - - | 0.1764±0.008 |
|   |    |  | (0.1656–0.1955) |
| 22 | RS1 | 2–4 | 2–3 |
| 23 | RS2 | 8–11 | 8–9 |
| 24 | RM | 18–23 | 18–22 |
| 25 | RR | 35–53 | 33–40 |

(*Continued*)

**Table 4.** (Continued)

| S. No. | Characters | *T. chilotraeae* (Male) | *T. chilotraeae* (Female) |
|---|---|---|---|
| 26 | NFH | 36–41 | - - - - - - - - - - - - |

Mean±standard deviation (top value) and range (parentheses); S. No. 1–21 are in mm and 22–26 are in numbers.

### Morphometric measurements of *T. danausicida*

The length of males in *T. danausicida* were quite longer than females. The HL, HW, EW and MS in both males and females were found to be in the same range (Table 8). Therefore, the result of the present study was supported by Nagaraja [23] & Nagaraja and Mohanraj [27]. The mean FL and FW in males were 0.1654±0.009 mm and 0.0370±0.003 mm, respectively. In males, the FWL was about 1.9930±0.023 times the width (FWW) whereas the same in females was 2.0083±0.003 (Table 9). Nagaraja [23] recorded the FWL/FWW (1.894) and MFW/FWW as (0.1386) for *T. danausicida*, which are in accordance with the results of present study. The ratio, FWW/MFW was observed equal for males and females in *T. danausicida*. Images for dissected characters are given in Figs 8 and 9. The ratio of HWW/MHW was higher in males than females. Similarly, in females, the OL was about 1.2067±0.043 times of the HTL. The similar kind of results was reported by Yousuf *et al.* [50].

### *Trichogramma dendrolimi* Matsumura, 1926: 45

**Diagnosis.** The colour in males is yellow with blackish abdomen and mesoscutum. Antennal hairs are tapered and long; length is equal to the width of flagellum. The fringes on fore wings are about 1/8th of fore wings width. In male genitalia, DEG is highly sclerotised with constriction at base, having broadly rounded lateral lobes with acute apex. CS is below the level of GF tip. MVP is broad at base; CR paired, extending and in a single ridge and about the margin of gonobase. Aedeagus as long as apodemes, both together shorter than hind tibial length. Female, yellow with abdominal terga as black. Antenna clubbed with few short hairs. Ovipositor is slightly longer or somewhat equal to the hind tibia [19, 41, 42, 51, 52].

### Morphometric measurements of *T. dendrolimi*

The BL was found longer in males than in females. The mean HL, HW and EW show almost equal in both males and females with little more shift toward males (Table 10). The study carried out by Chan and Chou [51] is in support with the result obtained in the present study. The mean ratio of FHL/FW, FL/HTL, OL/ACL and HTL/ACL were 2.1142±0.124, 0.9191 ±0.045, 2.5032±0.154 and 2.0431±0.103, respectively (Table 11). Nagarkatti and Nagaraja [19] reported the ratio of FHL and FW as 2.5, which is quite similar to the result of the present study. Similarly, this ratio (2–2.5) was reported by Viggiani and Laudonia [42]. The GCL was about 2.3564±0.135 times to its width (GCW) which ranged from 2.1034 to 2.56. Chiriac [41] have also provided a brief structure of male genitalia in *T. dendrolimi*. Images for dissected characters are presented in Figs 10 and 11.

### Morphometric comparison in males of *Trichogramma* spp.

In males, the body length (BL) is identified as an important character. The BL for *T. brassicae*, *T. chilotraeae* and *T. danaidiphaga* were less than 0.5022 whereas, *T. danausicida* and *T. dendrolimi* has BL more than 0.5115. Similarly, HL was mostly least for *T. brassicae* when compared with males of other four species of *Trichogramma*. The HW could be a vital character, as

*Trichogramma chilotraeae* ♂

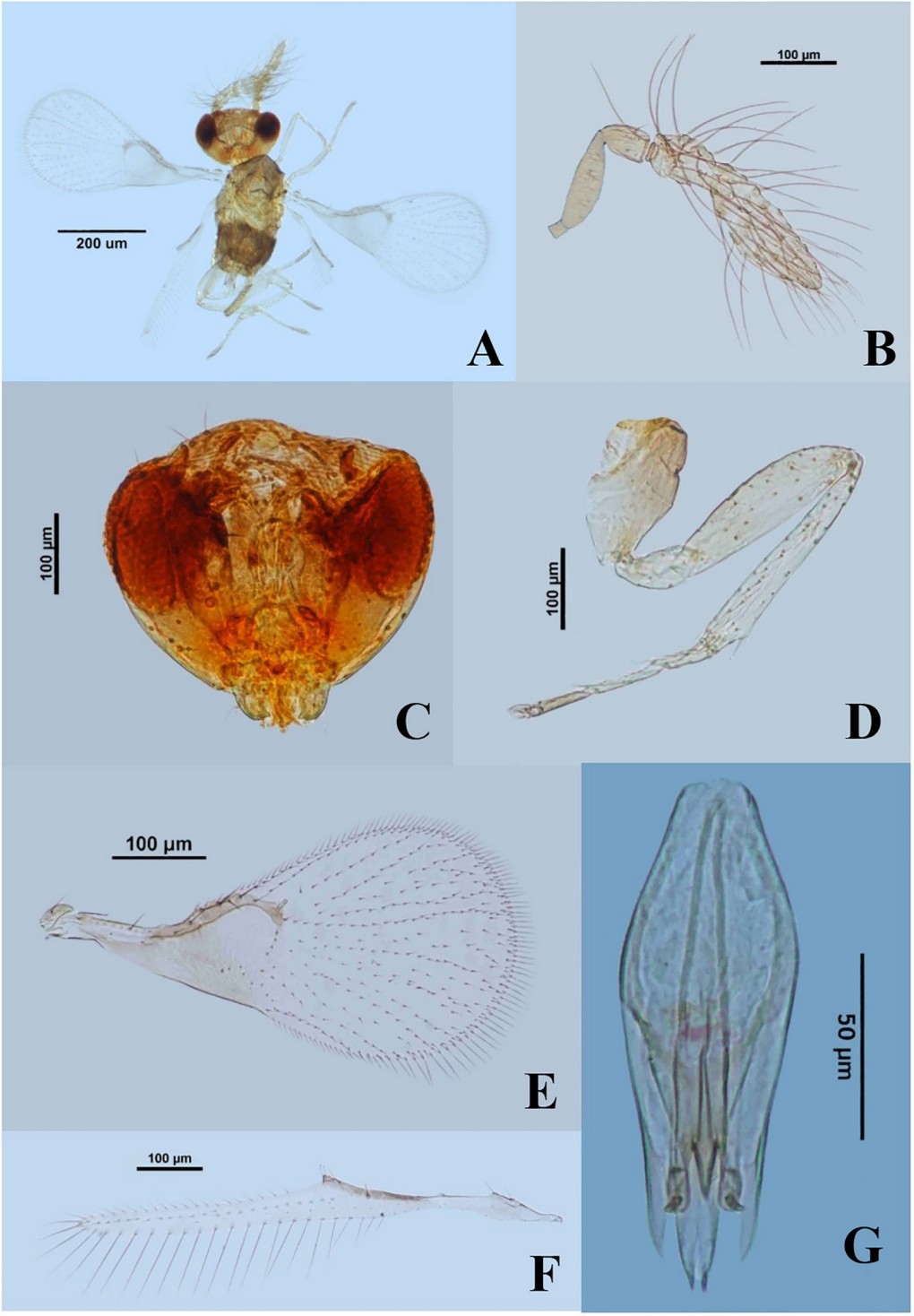

**Fig 4. Morphological characters of male *Trichogramma chilotraeae* Nagaraja and Nagarkatti.** A—whole body, B—antenna, C—head, D—hind tibia, E—fore wing, F—hind wing, G- genital capsule.

## *Trichogramma chilotraeae* ♀

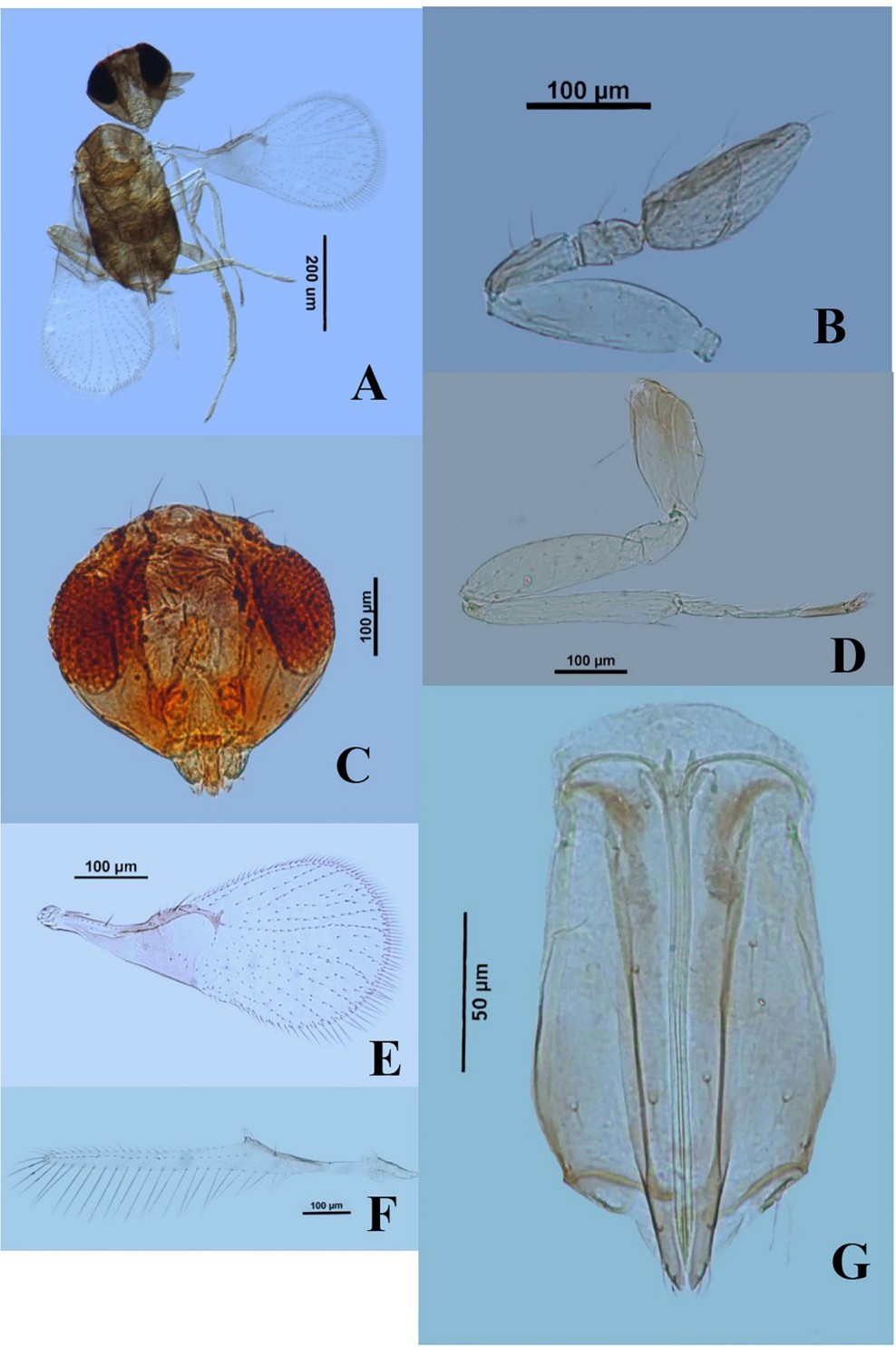

**Fig 5. Morphological characters of female *Trichogramma chilotraeae* Nagaraja and Nagarkatti.** A—whole body, B—antenna, C—head, D—hind tibia, E—fore wing, F—hind wing, G—ovipositor.

**Table 5. Morphometrics of ratios characters for males and females of *Trichogramma chilotraeae* Nagaraja and Nagarkatti.**

| S. No. | Characters | *T. chilotraeae* (Male) | *T. chilotraeae* (Female) |
|---|---|---|---|
| 1 | FHL/FW | 2.3269±0.313 | - - - - - - - - - - - |
| | | (2.0000–3.0000) | |
| 2 | FL/FW | 4.8167±0.743 | - - - - - - - - - - - |
| | | (4.0909–6.4286) | |
| 3 | FL/HTL | 0.9761±0.035 | - - - - - - - - - - - |
| | | (0.9333–1.0465) | |
| 4 | ACL/ACW | - - - - - - - - - - - | 2.3349±0.176 |
| | | | (2.0909–2.6250) |
| 5 | OL/ACL | - - - - - - - - - - - | 2.2290±0.112 |
| | | | (2.0725–2.4233) |
| 6 | HTL/ACL | - - - - - - - - - - - | 2.0625±0.092 |
| | | | (1.8696–2.1905) |
| 7 | FWL/FWW | 2.0836±0.024 | 2.0426±0.092 |
| | | (2.0370–2.1250) | (1.9200–2.2083) |
| 8 | FWW/MFW | 6.5267±0.374 | 6.4321±0.405 |
| | | (5.9463–7.2783) | (5.8125–7.2205) |
| 9 | HWW/MHW | 0.6350±0.026 | 0.6767±0.045 |
| | | (0.6154–0.6800) | (0.5769–0.7391) |
| 10 | GCL/GCW | 2.9777±0.090 | - - - - - - - - - - - |
| | | (2.8636–3.1429) | |
| 11 | GCL/HTL | 0.8617±0.041 | - - - - - - - - - - - |
| | | (0.8084–0.9627) | |
| 12 | OL/HTL | - - - - - - - - - - - | 1.0810±0.034 |
| | | | (1.0454–1.1590) |

Mean±standard deviation (top value) and range (parentheses).

Matsumura [32] used head characteristics to identify and separate the species of *Trichogramma*. In present study, *T. chilotraeae*, *T. danausicida* and *T. dendrolimi* can be differentiated from *T. brassicae* and *T. danaidiphaga* on the basis of morphometrics of characters. Character, EW shows dynamic change in almost all the male *Trichogramma* spp. *Trichogramma brassicae* and *T. danaidiphaga* have less EW length than others. Characters such as MS, FL and FW were not able to differentiate the species as there is very little changes in their values. FHL in *T. brassicae* was ranges from 0.0843 mm to 0.0990 mm, whereas, other four species show minimum flagellar hair length as 0.0733 mm (except for *T. dendrolimi*) and maximum as 0.0807 mm. Similar result was recorded by Nagaraja and Nagarkatti [34], Alba [47], Viggiani and Laudonia [42] & del Pino [45]. Character, FWL and FWW Individually does not show significant difference within the studied *Trichogramma* species. A value of 0.0368 mm for MFW in males was able to distinguish *T. danaidiphaga* from *T. brassicae* and *T. danausicida*. HWL in *T. dendrolimi* was ranged from 0.3910 mm to 0.4370 mm, whereas, *T. brassicae* and *T. danaidiphaga* have maximum length (0.3933 mm and 0.3956 mm, respectively). The character HTW in *T. dendrolimi* was showing highest value (0.0282 mm) with minimum and maximum values of 0.0529 mm and 0.0575 mm, respectively. Rest of the four species shows less wide hind tibia when compared to *T. dendrolimi*. *T. Danaidiphaga*, which has smallest genital capsule length (0.1267 mm) and width (0.0389 mm). Mean value of character CTG was observed in order of *T. danaidiphaga*<*T. dendrolimi*<*T. danausicida*<*T. chilotraeae*< *T. brassicae*. The aedeagus length (AL) was pointedly different in *T. chilotraeae* (0.1288–0.1472 mm)

**Table 6. Morphometrics of main characters for males and females of *Trichogramma danaidiphaga* Nagaraja and Prashanth.**

| S. No. | Characters | *T. danaidiphaga* (Male) | *T. danaidiphaga* (Female) |
|---|---|---|---|
| 1 | BL | 0.4790±0.015 | 0.5031±0.013 |
| | | (0.4557–0.4929) | (0.4836–0.5208) |
| 2 | HL | 0.1697±0.011 | 0.1725±0.014 |
| | | (0.1564–0.1909) | (0.1541–0.2024) |
| 3 | HW | 0.1888±0.014 | 0.1960±0.013 |
| | | (0.1725–0.2139) | (0.1817–0.2185) |
| 4 | EW | 0.0736±0.005 | 0.0826±0.005 |
| | | (0.0690–0.0828) | (0.0759–0.0897) |
| 5 | MS | 0.0529±0.003 | 0.0529±0.002 |
| | | (0.0483–0.0575) | (0.0506–0.0552) |
| 6 | FL/ACL | 0.1452±0.006 | 0.0807±0.003 |
| | | (0.1357–0.1577) | (0.0770–0.0880) |
| 7 | FW/ ACW | 0.0341±0.002 | 0.0363±0.002 |
| | | (0.0293–0.0367) | (0.0330–0.0403) |
| 8 | FHL | 0.0777±0.002 | - - - - - - - - - - - - |
| | | (0.0733–0.0807) | |
| 9 | FWL | 0.4715±0.024 | 0.4678±0.015 |
| | | (0.4464–0.5208) | (0.4464–0.4929) |
| 10 | FWW | 0.2381±0.010 | 0.2353±0.006 |
| | | (0.2232–0.2604) | (0.2232–0.2418) |
| 11 | MFW | 0.0435±0.005 | 0.0398±0.003 |
| | | (0.0368–0.0529) | (0.0368–0.0483) |
| 12 | HWL | 0.3758±0.010 | 0.3687±0.010 |
| | | (0.3611–0.3956) | (0.3496–0.3818) |
| 13 | HWW | 0.0396±0.001 | 0.0389±0.002 |
| | | (0.0391–0.0414) | (0.0368–0.0414) |
| 14 | MHW | 0.0545±0.003 | 0.0552±0.002 |
| | | (0.0483–0.0575) | (0.0529–0.0575) |
| 15 | HTL | 0.1595±0.006 | 0.1643±0.003 |
| | | (0.1503–0.1650) | (0.1577–0.1687) |
| 16 | HTW | 0.0242±0.002 | 0.0257±0.000 |
| | | (0.0220–0.0257) | (0.0257–0.0257) |
| 17 | GCL | 0.1267±0.004 | - - - - - - - - - - - - |
| | | (0.1173–0.1311) | |
| 18 | GCW | 0.0389±0.001 | - - - - - - - - - - - - |
| | | (0.0368–0.0391) | |
| 19 | CTG | 0.0074±0.001 | - - - - - - - - - - - - |
| | | (0.0069–0.0092) | |
| 20 | AL | 0.1244±0.004 | - - - - - - - - - - - - |
| | | (0.1150–0.1288) | |
| 21 | OL | - - - - - - - - - - - - | 0.1661±0.005 |
| | | | (0.1564–0.1725) |
| 22 | RS1 | 3–4 | 3–4 |
| 23 | RS2 | 9–11 | 9–11 |
| 24 | RM | 15–22 | 17–18 |
| 25 | RR | 28–37 | 29–37 |

(*Continued*)

**Table 6.** (Continued)

| S. No. | Characters | *T. danaidiphaga* (Male) | *T. danaidiphaga* (Female) |
|---|---|---|---|
| 26 | NFH | 30–34 | - - - - - - - - - - - - |

Mean±standard deviation (top value) and range (parentheses); S. No. 1–21 are in mm and 22–26 are in numbers.

and *T. danaidiphaga* (0.1150–0.1288 mm) which follows the results presented by similar result was presented by Nagaraja and Nagarkatti [34]; Ruiz and Korytkowski [49]; Viggiani and Laudonia [42]; Honda *et al.* [52] & Del Pino [45]. *Trichogramma dendrolimi* has highest number of setae on RS1 than other species. Similarly, setae between RS2 and RM was a unique character to for *T. brassicae* (38–45) and *T. danaidiphaga* (28–37).

The ratio of FHL and FW was least in *T. danausicida* (2.0914) and *T. dendrolimi* (2.1142) than other three species of *Trichogramma*. This result was supported by Viggiani and Laudonia [42] & Chiriac [41]. Similarly, the ratio of FWL and FWW in males of *T. danausicida*, *T. danaidiphaga* and *T. dendrolimi* was lesser (<0.19) than *T. brassicae* and *T. chilotraeae* which confirms the morphology mentioned by Alba [47] & Del Pino [45]. Also, the ratio of HWW and MHW was able to differentiate *T. chilotraeae* (0.6154–0.6800) from *T. danaidiphaga* (0.6800–0.8571). The GCL/GCW was least in *T. dendrolimi* (2.3564) than rest of the species. Some of the above-mentioned ratio characters were also used by Nagaraja and Nagarkatti [34],

**Table 7. Morphometrics of ratios characters for males and females of *Trichogramma danaidiphaga* Nagaraja and Prashanth.**

| S. No. | Characters | *T. danaidiphaga* (Male) | *T. danaidiphaga* (Female) |
|---|---|---|---|
| 1 | FHL/FW | 2.2903±0.178 | - - - - - - - - - - - - |
|  |  | (2.1000–2.6250) |  |
| 2 | FL/FW | 4.2750±0.303 | - - - - - - - - - - - - |
|  |  | (3.7000–4.7500) |  |
| 3 | FL/HTL | 0.9117±0.054 | - - - - - - - - - - - - |
|  |  | (0.8222–0.9762) |  |
| 4 | ACL/ACW | - - - - - - - - - - - - | 2.2260±0.106 |
|  |  |  | (2.1000–2.4444) |
| 5 | OL/ACL | - - - - - - - - - - - - | 2.0609±0.086 |
|  |  |  | (1.9387–2.1803) |
| 6 | HTL/ACL | - - - - - - - - - - - - | 2.0393±0.084 |
|  |  |  | (1.9167–2.1429) |
| 7 | FWL/FWW | 1.9805±0.053 | 1.9885±0.052 |
|  |  | (1.8846–2.0400) | (1.8846–2.0800) |
| 8 | FWW/MFW | 5.5587±0.791 | 5.9413±0.413 |
|  |  | (4.3951–7.0761) | (5.0062–6.5706) |
| 9 | HWW/MHW | 0.7293±0.064 | 0.7049±0.039 |
|  |  | (0.6800–0.8571) | (0.6400–0.7826) |
| 10 | GCL/GCW | 3.2621±0.147 | - - - - - - - - - - - - |
|  |  | (3.0000–3.5625) |  |
| 11 | GCL/HTL | 0.7950±0.028 | - - - - - - - - - - - - |
|  |  | (0.7527–0.8512) |  |
| 12 | OL/HTL | - - - - - - - - - - - - | 1.0109±0.030 |
|  |  |  | (0.9617–1.0648) |

Mean±standard deviation (top value) and range (parentheses).

*Trichogramma danaidiphaga* ♂

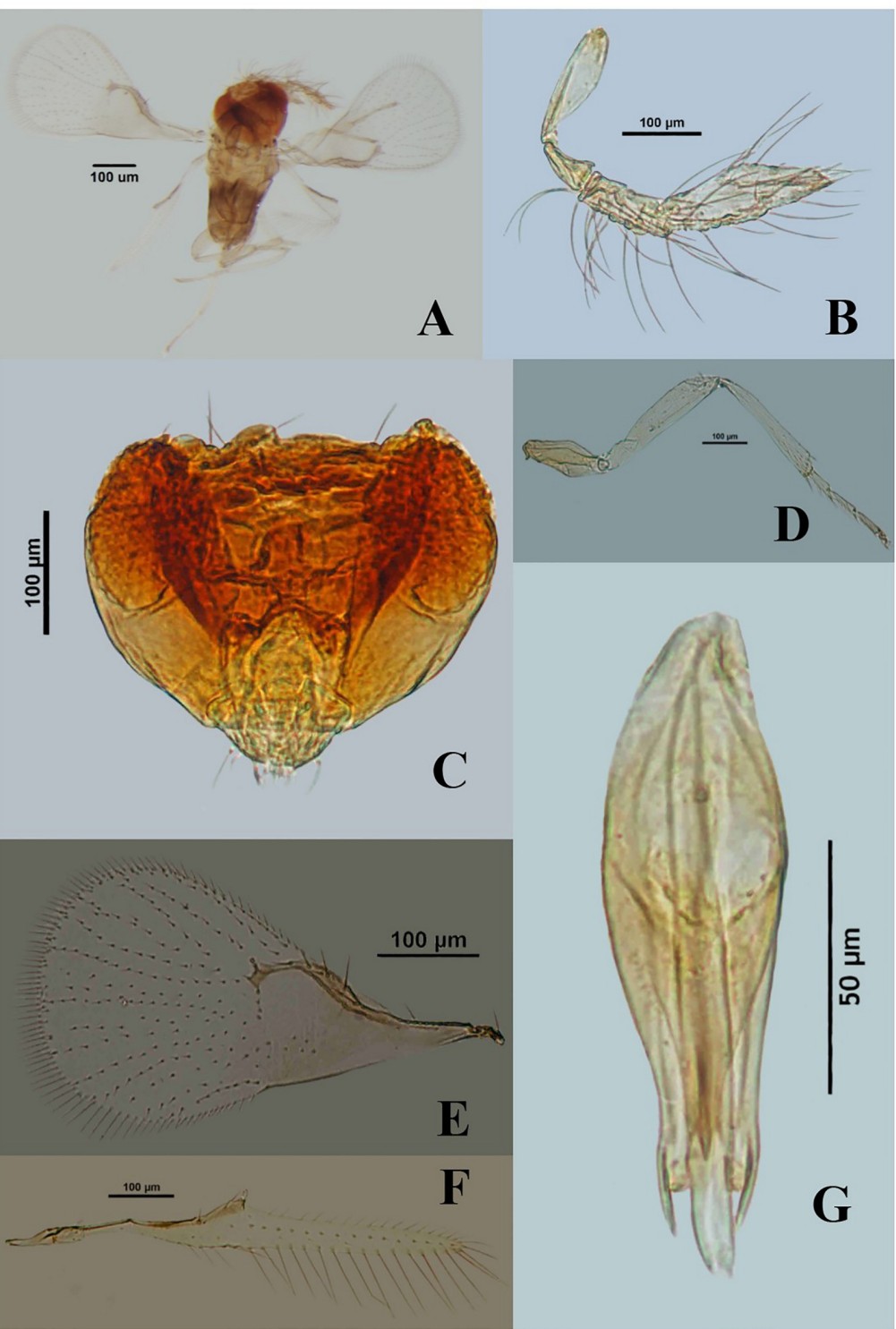

**Fig 6. Morphological characters of male *Trichogramma danaidiphaga* Nagaraja and Prashanth.** A—whole body, B—antenna, C—head, D—hind tibia, E—fore wing, F—hind wing, G—genital capsule.

*Trichogramma danaidiphaga* ♀

**Fig 7. Morphological characters of female *Trichogramma danaidiphaga* Nagaraja and Prashanth.** A—whole body, B—antenna, C—head, D—hind tibia, E—fore wing, F—hind wing, G—ovipositor.

**Table 8. Morphometrics of main characters for males and females of *Trichogramma danausicida* Nagaraja.**

| S. No. | Characters | *T. danausicida* (Male) | *T. danausicida* (Female) |
|---|---|---|---|
| 1 | BL | 0.5320±0.011 | 0.5915±0.024 |
| | | (0.5115–0.5487) | (0.5487–0.6324) |
| 2 | HL | 0.1962±0.016 | 0.1950±0.009 |
| | | (0.1748–0.2185) | (0.1840–0.2116) |
| 3 | HW | 0.2160±0.015 | 0.2164±0.007 |
| | | (0.2001–0.2369) | (0.2047–0.2277) |
| 4 | EW | 0.0945±0.004 | 0.0964±0.003 |
| | | (0.0897–0.1012) | (0.0920–0.1012) |
| 5 | MS | 0.0547±0.002 | 0.0534±0.002 |
| | | (0.0506–0.0575) | (0.0506–0.0575) |
| 6 | FL/ACL | 0.1654±0.009 | 0.0821±0.003 |
| | | (0.1503–0.1760) | (0.0770–0.0880) |
| 7 | FW/ ACW | 0.0370±0.003 | 0.0363±0.002 |
| | | (0.0330–0.0403) | (0.0330–0.0403) |
| 8 | FHL | 0.0770±0.002 | - - - - - - - - - - - |
| | | (0.0733–0.0807) | |
| 9 | FWL | 0.5115±0.023 | 0.4743±0.011 |
| | | (0.4650–0.5394) | (0.4650–0.4929) |
| 10 | FWW | 0.2567±0.012 | 0.2362±0.007 |
| | | (0.2325–0.2697) | (0.2325–0.2511) |
| 11 | MFW | 0.0357±0.002 | 0.0327±0.001 |
| | | (0.0322–0.0368) | (0.0299–0.0345) |
| 12 | HWL | 0.3974±0.007 | 0.3887±0.010 |
| | | (0.3841–0.4071) | (0.3795–0.4094) |
| 13 | HWW | 0.0398±0.003 | 0.0384±0.002 |
| | | (0.0368–0.0437) | (0.0345–0.0414) |
| 14 | MHW | 0.0564±0.002 | 0.0534±0.002 |
| | | (0.0529–0.0598) | (0.0506–0.0575) |
| 15 | HTL | 0.1672±0.006 | 0.1654±0.006 |
| | | (0.1577–0.1760) | (0.1540–0.1723) |
| 16 | HTW | 0.0253±0.001 | 0.0260±0.001 |
| | | (0.0220–0.0257) | (0.0257–0.0293) |
| 17 | GCL | 0.1449±0.004 | - - - - - - - - - - - |
| | | (0.1403–0.1518) | |
| 18 | GCW | 0.0439±0.002 | - - - - - - - - - - - |
| | | (0.0414–0.0483) | |
| 19 | CTG | 0.0097±0.001 | - - - - - - - - - - - |
| | | (0.0092–0.0115) | |
| 20 | AL | 0.1325±0.004 | - - - - - - - - - - - |
| | | (0.1265–0.1403) | |
| 21 | OL | - - - - - - - - - - - | 0.1994±0.006 |
| | | | (0.1909–0.2070) |
| 22 | RS1 | 2–4 | 2–3 |
| 23 | RS2 | 8–11 | 8–11 |
| 24 | RM | 17–22 | 16–26 |
| 25 | RR | 32–43 | 33–39 |

(*Continued*)

**Table 8.** (Continued)

| S. No. | Characters | T. danausicida (Male) | T. danausicida (Female) |
|--------|-----------|----------------------|------------------------|
| 26 | NFH | 32–39 | - - - - - - - - - - - - |

Mean±standard deviation (top value) and range (parentheses); S. No. 1–21 are in mm and 22–26 are in numbers.

Nagarkatti and Nagaraja [19], Ruiz and Korytkowski [49], Chan and Chou [51] & Nagaraja and Mohanraj [27].

## Morphometric comparison in females of *Trichogramma* spp.

The BL in female of *T. danausicida* (0.5915 mm) was significantly longer than other. The variations of BL within the specimens of *T. brassicae*, *T. chilotraeae*, *T. danaidiphaga* and *T. dendrolimi* were far less than the *T. danausicida* as also discussed by Nagaraja and Mohanraj [27, 28]. The mean HL in females *Trichogramma* were found in the order of *T. danaidiphaga*<*T. brassicae*<*T. dendrolimi*<*T. chilotraeae*<*T. danausicida*. The character, EW was able to separate *T. danaidiphaga* (0.0759–0.0897 mm) from *T. danausicida* (0.0920–0.1012 mm) and *T. dendrolimi* (0.0897–0.0966 mm). The character, ACL was least in *T. dendrolimi* (0.0752 mm) followed by *T. chilotraeae* (0.0792 mm). In *T. brassicae*, the variation in ACW was high and

**Table 9.** Morphometrics of ratios characters for males and females of *Trichogramma danausicida* Nagaraja.

| S. No. | Characters | T. danausicida (Male) | T. danausicida (Female) |
|--------|-----------|----------------------|------------------------|
| 1 | FHL/FW | 2.0914±0.196 | - - - - - - - - - - - - |
|   |   | (1.8182–2.4444) |   |
| 2 | FL/FW | 4.4855±0.396 | - - - - - - - - - - - - |
|   |   | (4.0909–5.2222) |   |
| 3 | FL/HTL | 0.9892±0.047 | - - - - - - - - - - - - |
|   |   | (0.9130–1.0909) |   |
| 4 | ACL/ACW | - - - - - - - - - - - - | 2.2702±0.173 |
|   |   |   | (2.0909–2.6667) |
| 5 | OL/ACL | - - - - - - - - - - - - | 2.4311±0.125 |
|   |   |   | (2.2634–2.6881) |
| 6 | HTL/ACL | - - - - - - - - - - - - | 2.0162±0.108 |
|   |   |   | (1.8261–2.1905) |
| 7 | FWL/FWW | 1.9930±0.023 | 2.0083±0.003 |
|   |   | (1.9643–2.0370) | (1.9615–2.0400) |
| 8 | FWW/MFW | 7.2142±0.483 | 7.2433±0.326 |
|   |   | (6.5706–8.0869) | (6.7391–7.7981) |
| 9 | HWW/MHW | 0.7069±0.056 | 0.7216±0.063 |
|   |   | (0.6400–0.7826) | (0.6522–0.8182) |
| 10 | GCL/GCW | 3.3058±0.177 | - - - - - - - - - - - - |
|   |   | (3.0476–3.5556) |   |
| 11 | GCL/HTL | 0.8672±0.030 | - - - - - - - - - - - - |
|   |   | (0.8101–0.9123) |   |
| 12 | OL/HTL | - - - - - - - - - - - - | 1.2067±0.043 |
|   |   |   | (1.1317–1.2694) |

Mean±standard deviation (top value) and range (parentheses).

*Trichogramma danausicida* ♂

**Fig 8. Morphological characters of male *Trichogramma danausicida* Nagaraja.** A- whole body, B- antenna, C- head, D-hind tibia, E- fore wing, F- hind wing, G- genital capsule.

*Trichogramma danausicida* ♀

**Fig 9. Morphological characters of female *Trichogramma danausicida* Nagaraja.** A- whole body, B- antenna, C-head, D- hind tibia, E- fore wing, F- hind wing, G- ovipositor.

**Table 10. Morphometrics of main characters for males and females of *Trichogramma dendrolimi* Matsumura.**

| S. No. | Characters | *T. dendrolimi* (Male) | *T. dendrolimi* (Female) |
|--------|------------|------------------------|--------------------------|
| 1 | BL | 0.5292±0.009 | 0.4948±0.009 |
| | | (0.5115–0.5394) | (0.4836–0.5115) |
| 2 | HL | 0.1907±0.007 | 0.1835±0.009 |
| | | (0.1794–0.2001) | (0.1725–0.1955) |
| 3 | HW | 0.2116±0.011 | 0.2024±0.009 |
| | | (0.1978–0.2277) | (0.1886–0.2162) |
| 4 | EW | 0.0934±0.005 | 0.0932±0.002 |
| | | (0.0851–0.1012) | (0.0897–0.0966) |
| 5 | MS | 0.0550±0.003 | 0.0513±0.003 |
| | | (0.0506–0.0575) | (0.0483–0.0575) |
| 6 | FL/ACL | 0.1496±0.007 | 0.0752±0.004 |
| | | (0.1430–0.1650) | (0.0660–0.0807) |
| 7 | FW/ ACW | 0.0374±0.002 | 0.0348±0.002 |
| | | (0.0330–0.0403) | (0.0330–0.0367) |
| 8 | FHL | 0.0788±0.002 | - - - - - - - - - - - - |
| | | (0.0770–0.0807) | |
| 9 | FWL | 0.5115±0.020 | 0.4883±0.013 |
| | | (0.4836–0.5394) | (0.4743–0.5115) |
| 10 | FWW | 0.2576±0.012 | 0.2446±0.008 |
| | | (0.2418–0.2790) | (0.2325–0.2511) |
| 11 | MFW | 0.0363±0.002 | 0.0354±0.002 |
| | | (0.0322–0.0391) | (0.0322–0.0368) |
| 12 | HWL | 0.4115±0.014 | 0.3885±0.008 |
| | | (0.3910–0.4370) | (0.3795–0.4025) |
| 13 | HWW | 0.0414±0.004 | 0.0375±0.001 |
| | | (0.0368–0.0460) | (0.0368–0.0391) |
| 14 | MHW | 0.0568±0.002 | 0.0552±0.001 |
| | | (0.0529–0.0575) | (0.0529–0.0575) |
| 15 | HTL | 0.1628±0.004 | 0.1533±0.005 |
| | | (0.1577–0.1687) | (0.1467–0.1613) |
| 16 | HTW | 0.0282±0.002 | 0.0257±0.000 |
| | | (0.0257–0.0330) | (0.0257–0.0257) |
| 17 | GCL | 0.1461±0.005 | - - - - - - - - - - - - |
| | | (0.1403–0.1541) | |
| 18 | GCW | 0.0621±0.003 | - - - - - - - - - - - - |
| | | (0.0575–0.0667) | |
| 19 | CTG | 0.0083±0.002 | - - - - - - - - - - - - |
| | | (0.0069–0.0115) | |
| 20 | AL | 0.1417±0.006 | - - - - - - - - - - - - |
| | | (0.1334–0.1495) | |
| 21 | OL | - - - - - - - - - - - - | 0.1877±0.004 |
| | | | (0.1817–0.1955) |
| 22 | RS1 | 4–6 | 4–6 |
| 23 | RS2 | 7–11 | 9–11 |
| 24 | RM | 17–21 | 17–22 |
| 25 | RR | 30–38 | 34–43 |

(*Continued*)

**Table 10.** (Continued)

| S. No. | Characters | *T. dendrolimi* (Male) | *T. dendrolimi* (Female) |
|--------|-----------|------------------------|--------------------------|
| 26 | NFH | 32–36 | - - - - - - - - - - - |

Mean±standard deviation (top value) and range (parentheses); S. No. 1–21 are in mm and 22–26 are in numbers.

ranging from 0.0293 mm to 0.0403 mm though alone, it does not make better contribution in morphometrics studies species. Forewings characters such as FWL and FWW were not able to take for segregating the *Trichogramma* spp. except MFW. In T. *danausicida* and *T. dendrolimi*, MFW was recorded maximum with length of 0.0345 mm and 0.0322 mm respectively whereas, for *T. brassicae* the minimum length was 0.0368 mm to 0.0483 mm. This follows the results obtained by Nagaraja and Nagarkatti [34]; Nagaraja and Mohanraj [27, 28]. Similarly, the characters of hind wings such as HWL and HWW were not found useful to separate the different species when considered individually, but MHW was able to distinct most of *T. brassicae* from other four species being longest with exception in few specimens. It was observed that HTW is an important character for female *Trichogramma* spp. morphometry. HTW was measured as constant 0.0257 mm in two species (*T. danaidiphaga* and *T. dendrolimi*) as all the specimens were recorded with same width, whereas, minimum and maximum width of HTW in *T. danausicida* was measured as 0.0257 mm and 0.0293 mm, respectively. Majority of the

**Table 11. Morphometrics of ratios characters for males and females of *Trichogramma dendrolimi* Matsumura.**

| S. No. | Characters | *T. dendrolimi* (Male) | *T. dendrolimi* (Female) |
|--------|-----------|------------------------|--------------------------|
| 1 | FHL/FW | 2.1142±0.124 | - - - - - - - - - - - |
|   |          | (1.9091–2.3333) |  |
| 2 | FL/FW | 4.0115±0.279 | - - - - - - - - - - - |
|   |       | (3.6364–4.5000) |  |
| 3 | FL/HTL | 0.9191±0.045 | - - - - - - - - - - - |
|   |        | (0.8667–0.9783) |  |
| 4 | ACL/ACW | - - - - - - - - - - - | 2.1644±0.177 |
|   |         |  | (2.0000–2.4444) |
| 5 | OL/ACL | - - - - - - - - - - - | 2.5032±0.154 |
|   |        |  | (2.3663–2.8922) |
| 6 | HTL/ACL | - - - - - - - - - - - | 2.0431±0.103 |
|   |         |  | (1.8636–2.2222) |
| 7 | FWL/FWW | 1.9866±0.034 | 1.9967±0.038 |
|   |         | (1.9310–2.0385) | (1.9615–2.0400) |
| 8 | FWW/MFW | 7.1192±0.661 | 6.9153±0.312 |
|   |         | (6.4220–8.3758) | (6.3179–7.2783) |
| 9 | HWW/MHW | 0.7296±0.081 | 0.6794±0.024 |
|   |         | (0.6400–0.8696) | (0.6400–0.7083) |
| 10 | GCL/GCW | 2.3564±0.135 | - - - - - - - - - - - |
|    |         | (2.1034–2.5600) |  |
| 11 | GCL/HTL | 0.8969±0.023 | - - - - - - - - - - - |
|    |         | (0.8502–0.9199) |  |
| 12 | OL/HTL | - - - - - - - - - - - | 1.2255±0.048 |
|    |        |  | (1.1546–1.3015) |

Mean±standard deviation (top value) and range (parentheses).

**Fig 10. Morphological characters of male *Trichogramma dendrolimi* Matsumura.** A- whole body, B- antenna, C-head, D- hind tibia, E- fore wing, F- hind wing, G- genital capsule.

## *Trichogramma dendrolimi* ♀

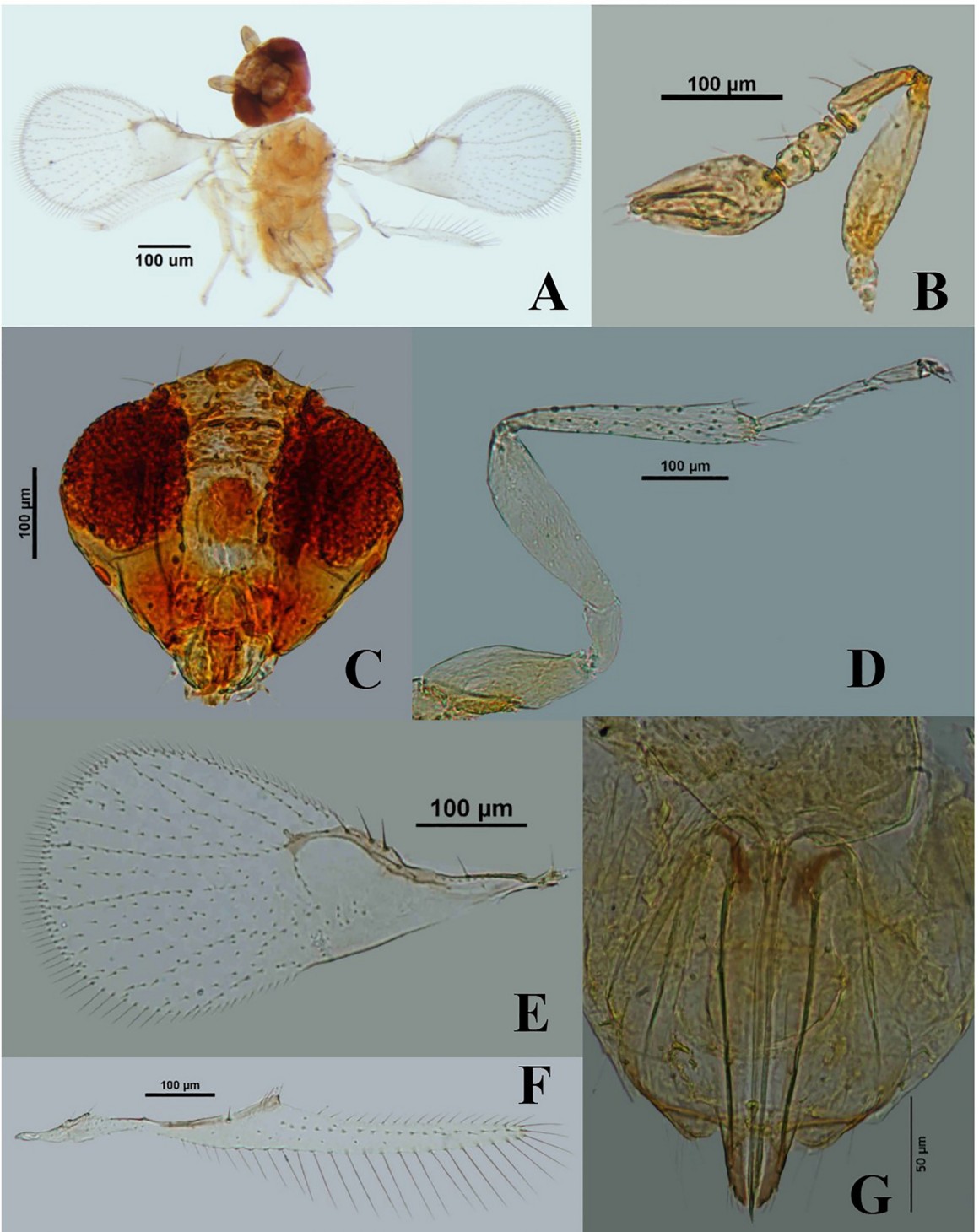

**Fig 11. Morphological characters of female *Trichogramma dendrolimi* Matsumura.** A- whole body, B- antennae, C- head, D- hind tibia, E- fore wing, F- hind wing, G- ovipositor.

species of female *Trichogramma* spp. was differentiated by ovipositor length as suggested by Matsumura [32] & Nagaraja and Nagarkatti [34]. In present study, shortest ovipositor was observed in *T. danaidiphaga*, while, longest was in *T. danausicida*. Also, females of *T. dendrolimi* can be separated from *T. danaidiphaga* by measuring their OL. The measurement of OL was in order of *T. danaidiphaga* (0.1661 mm)<*T. brassicae* (0.1727 mm)<*T. chilotraeae* (0.1764 mm)<*T. dendrolimi* (0.1877 mm)<*T. danausicida* (0.1994 mm). Number of setae in RS1 was maximum in *T. dendrolimi* (4–6) which differentiate this species from *T. danaidiphaga* (3–4) and *T. danausicida* (2–3). Also, setae in RM in *T. danaidiphaga* was counted b/w 17–18 whereas the same for *T. brassicae* and *T. chilotraeae* (18–24 and 18–22, respectively). The ratio of character OL and ACL was able to separate *T. danaidiphaga* (2.0609) from *T. danausicida* (2.4311) and *T. dendrolimi* (2.5032). Also, *T. danausicida* and *T. dendrolimi* can be differentiated from each other using OL/ACL. Similar, ratio was used by Nagaraja and Nagarkatti [34], Ruiz and Korytkowski [49] & Nagaraja and Mohanraj [27, 28] to identify and differentiate the *T. danaidiphaga* and *T. danausicida*. In females, the ratio, FWL/FWW in *T. danausicida* and *T. danaidiphaga* has recorded minimum value and may utilize to differentiate *T. brassicae* which has mean of 2.1712. Similarly, ratio of FWW/MFW was able to separate *T. danaidiphaga* (5.9413) and *T. danausicida* (7.2433) from other species. The ratio, OL/HTL was very important in separating *T. chilotraeae* from *T. dendrolimi*, *T. danaidiphaga* and *T. brassicae* from *T. danausicida* and *T. dendrolimi*. The increasing order of ratio OL/HTL was recorded as *T. danaidiphaga*<*T. brassicae*<*T. chilotraeae*<*T. danausicida*<*T. dendrolimi*. The result obtained for *T. chilotraeae* was in support of earlier work of Alba [47].

## Principal Component Analysis (PCA) of morphometric observations

The result of principal component analysis (PCA) for males of *Trichogramma* spp. depict that there are many principal components (PCs), of which four PCs are identified as major one as represented in Fig 12. The first four components are explaining the most significant contribution and rest produce the least contribution to the Eigenvalues. So, the major emphasis was given to the first four components in males of *Trichogramma* spp. Also, all the species were plotted based on their first (PC1) and second (PC2) component values in Fig 13. As per Fig 13, the first component of the morphometric dataset of males can differentiate *T. danaidiphaga* and *T. brassicae* from rest species of *Trichogramma* (*T. chilotraeae*, *T. danausicida* and *T. dendrolimi*). Similarly, the second component of PCA can separate majority of the *T. brassicae* and *T. chilotraeae* from rest of the *Trichogramma* spp.

The PCA analysis of females of *Trichogramma* spp. shows that there are three major principal components which has high Eigenvalues (Fig 14) and the rest of other PCs are participating less contribution to the Eigenvalues. The score plot (Fig 15) of PC1 and PC2 for females of *Trichogramma* spp. represent that the first component able to differentiate the majority of *T. dendrolimi* and *T. danausicida* from rest three species of *Trichogramma* (*T. danaidiphaga*, *T. brassicae* and *T. chilotraeae*). Also, second component is useful in differentiating *T. danaidiphaga* and *T. dendrolimi* from *T. danausicida* and *T. brassicae*. The first three-component represent greater amount of variability but first two are able to provide considerable information for all species studied. So, we considered first two-component axes for further analysis neglecting third component as it represents considerably less variability in the distribution in males.

Table 12 express the eigenvalues of PCA axes in both males and females. A total of 33 PCs in males and 27 PCs in females were studied for this purpose (Table 12). In males, the first component shows eigenvalues of 8.827 with the variability of more than 26%. In females, the first component shows eigenvalues of 6.785 with the variability of 25%. The first and second component together give variability of more than 48% and nearly 40% in both males and

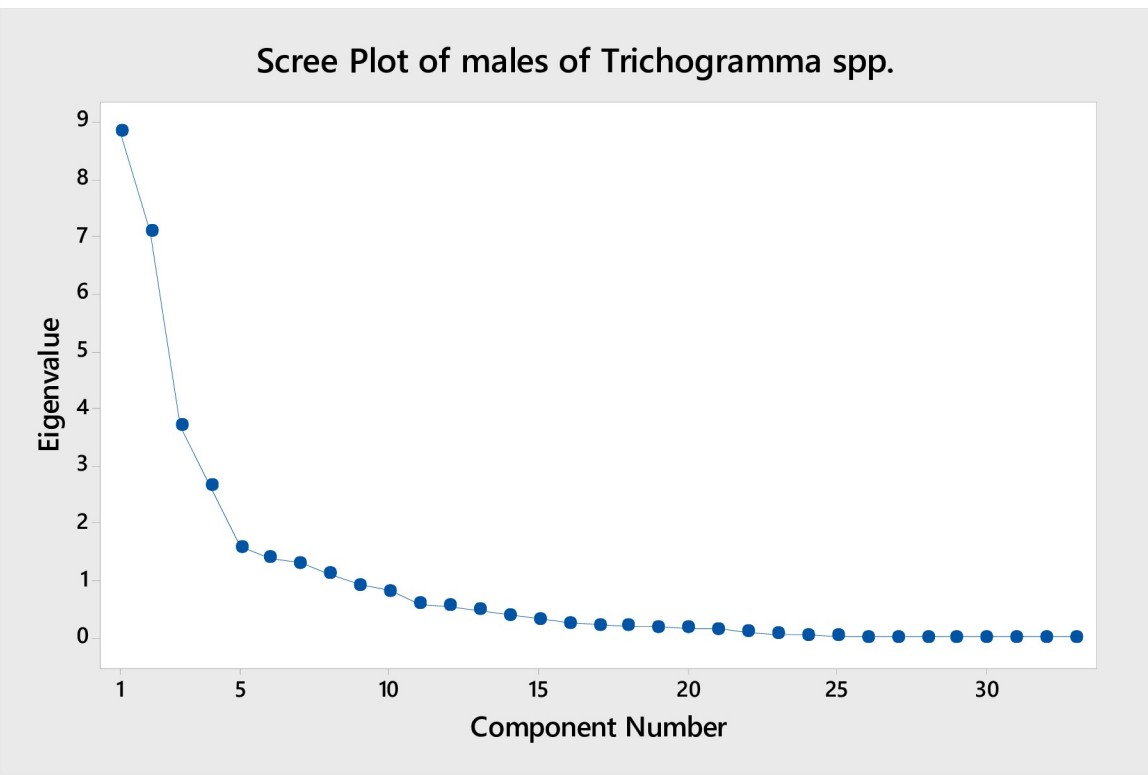

**Fig 12. Graph showing the components of male *Trichogramma* spp.**

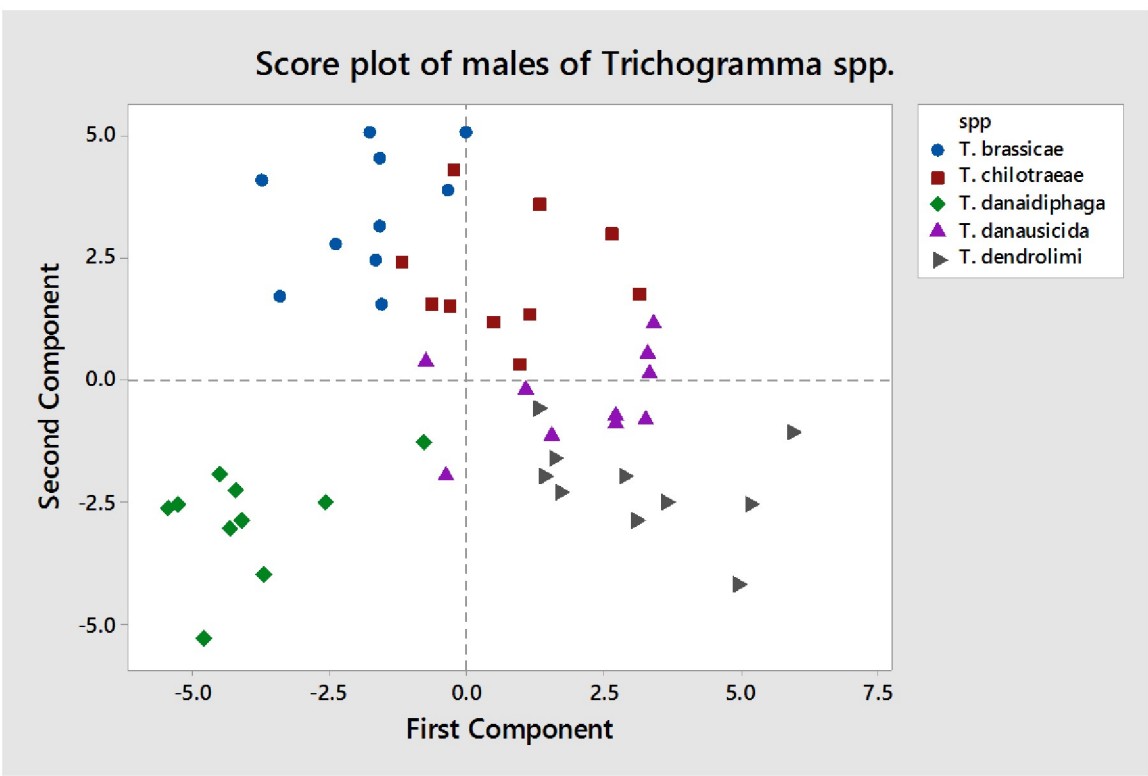

**Fig 13. Score plots of males of *Trichogramma* spp. in PC1 and PC2.**

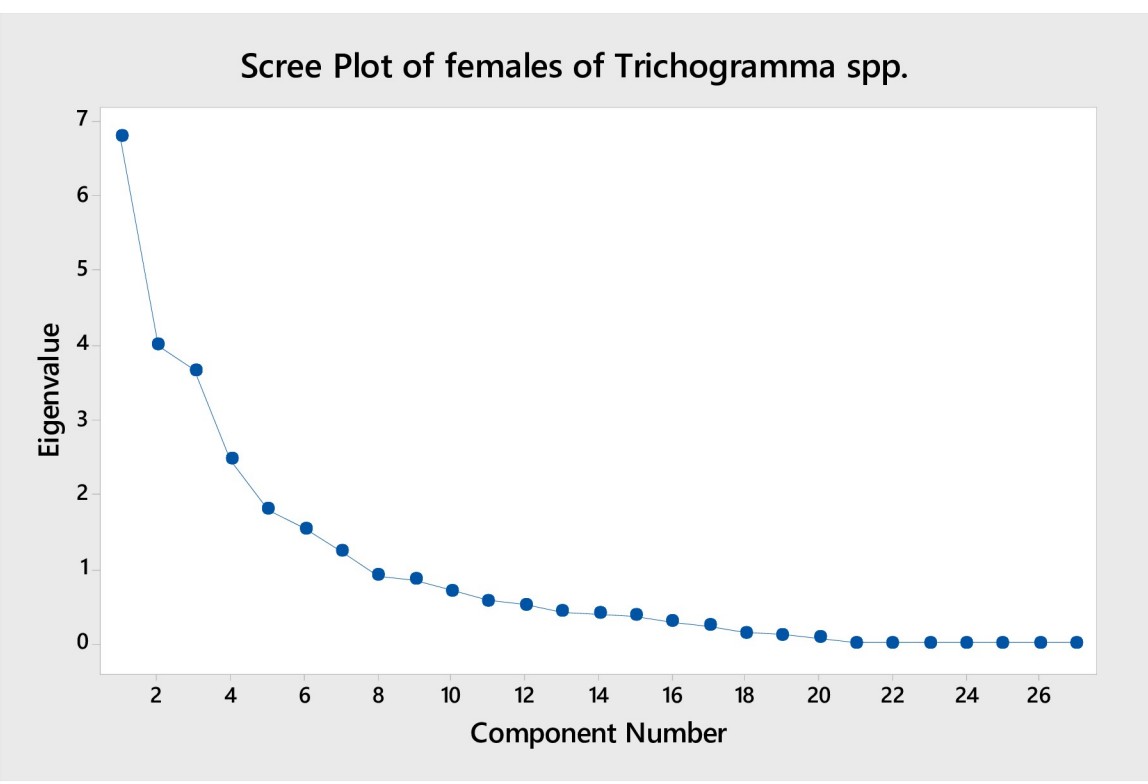

**Fig 14. Graph showing the components of female *Trichogramma* spp.**

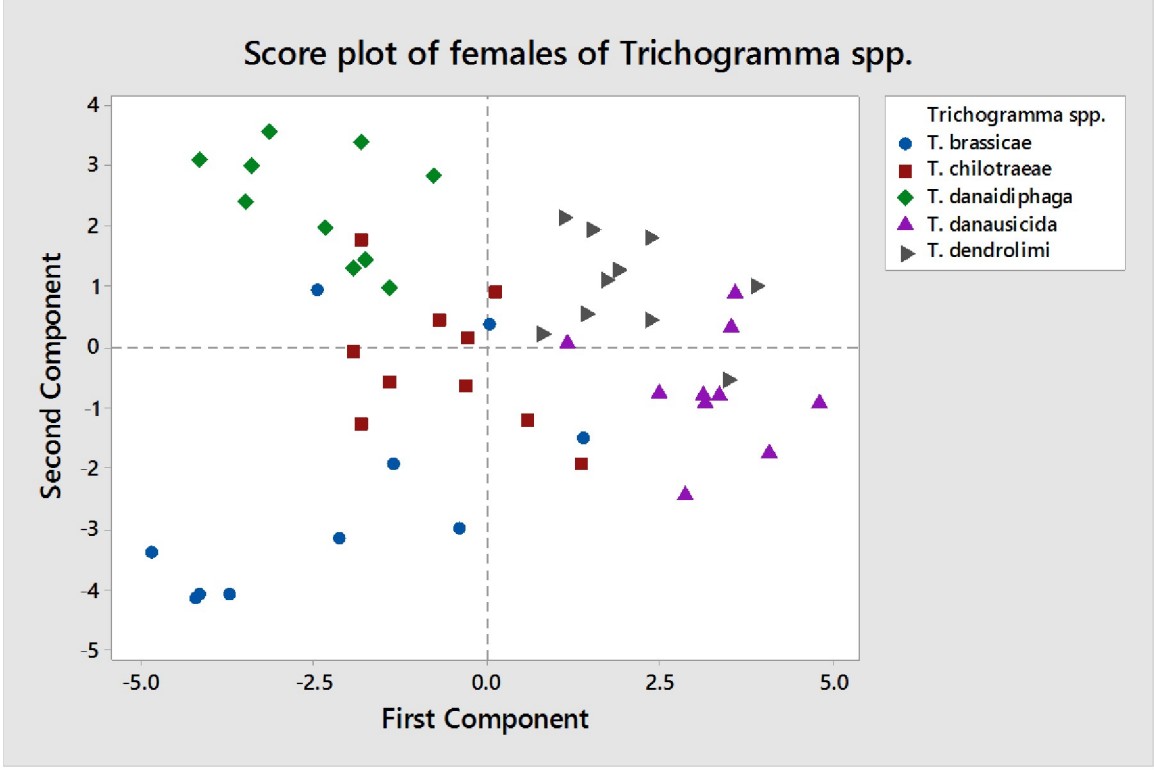

**Fig 15. Score plots of females of *Trichogramma* spp. in PC1 and PC2.**

**Table 12. Eigenvalues at different axes in *Trichogramma* spp.**

| Axes | Males | | | Females | | |
|---|---|---|---|---|---|---|
| | Eigenvalue | Variability (%) | Cumulative % | Eigenvalue | Variability (%) | Cumulative % |
| PC1 | 8.827 | 26.749 | 26.749 | 6.785 | 25.130 | 25.130 |
| PC2 | 7.069 | 21.420 | 48.169 | 3.988 | 14.769 | 39.899 |
| PC3 | 3.687 | 11.173 | 59.342 | 3.659 | 13.552 | 53.451 |
| PC4 | 2.647 | 8.020 | 67.362 | 2.466 | 9.132 | 62.583 |
| PC5 | 1.578 | 4.781 | 72.143 | 1.789 | 6.625 | 69.208 |
| PC6 | 1.389 | 4.210 | 76.353 | 1.532 | 5.674 | 74.882 |
| PC7 | 1.299 | 3.938 | 80.291 | 1.225 | 4.536 | 79.418 |
| PC8 | 1.108 | 3.356 | 83.647 | 0.899 | 3.329 | 82.747 |
| PC9 | 0.903 | 2.736 | 86.383 | 0.855 | 3.166 | 85.912 |
| PC10 | 0.812 | 2.462 | 88.845 | 0.697 | 2.580 | 88.492 |
| PC11 | 0.585 | 1.772 | 90.617 | 0.572 | 2.120 | 90.612 |
| PC12 | 0.547 | 1.659 | 92.276 | 0.513 | 1.902 | 92.514 |
| PC13 | 0.482 | 1.460 | 93.736 | 0.416 | 1.540 | 94.054 |
| PC14 | 0.391 | 1.185 | 94.921 | 0.387 | 1.432 | 95.486 |
| PC15 | 0.323 | 0.980 | 95.901 | 0.363 | 1.344 | 96.830 |
| PC16 | 0.256 | 0.776 | 96.676 | 0.287 | 1.062 | 97.892 |
| PC17 | 0.207 | 0.626 | 97.302 | 0.235 | 0.871 | 98.763 |
| PC18 | 0.194 | 0.587 | 97.889 | 0.143 | 0.529 | 99.292 |
| PC19 | 0.179 | 0.542 | 98.431 | 0.103 | 0.383 | 99.675 |
| PC20 | 0.154 | 0.468 | 98.899 | 0.076 | 0.281 | 99.957 |
| PC21 | 0.137 | 0.414 | 99.313 | 0.005 | 0.017 | 99.973 |
| PC22 | 0.092 | 0.280 | 99.593 | 0.003 | 0.013 | 99.986 |
| PC23 | 0.059 | 0.179 | 99.772 | 0.002 | 0.006 | 99.993 |
| PC24 | 0.031 | 0.095 | 99.867 | 0.001 | 0.003 | 99.996 |
| PC25 | 0.027 | 0.082 | 99.948 | 0.001 | 0.003 | 99.998 |
| PC26 | 0.007 | 0.022 | 99.970 | 0.000 | 0.001 | 99.999 |
| PC27 | 0.006 | 0.017 | 99.987 | 0.000 | 0.001 | 100.000 |
| PC28 | 0.002 | 0.007 | 99.994 | | | |
| PC29 | 0.001 | 0.003 | 99.998 | | | |
| PC30 | 0.001 | 0.002 | 99.999 | | | |
| PC31 | 0.000 | 0.000 | 100.000 | | | |
| PC32 | 0.000 | 0.000 | 100.000 | | | |
| PC33 | 0.000 | 0.000 | 100.000 | | | |

females, respectively. It means, the first two components in both males and females can explain and cover almost half of the variation among the species.

Tables 13 and 14 are explaining eigenvectors of observational character in both males and females, respectively for first 10 PCs. Eigenvectors shown in Table 13 characterize that the first principal component (PC1) in males are positively (weak) correlated with EW (Eye Width) and GCL (Length of Genitalia Capsule) because these characters show top values ≥0.290. Similarly, the second principal component (PC2) is positively (weak) associated with CTG (distance between the chelate structure and gonoforceps) and FL/FW (ratio of Flagellar length and width) with ≥0.300. Though, these association are not significantly more contributing but few characters together can be useful in establishing good correlation between the variables and PCs. Characters like FW (Flagellum width), RS1 (Number of setae in RS1) and HWW/MHW

**Table 13. Eigenvectors of PCA for the males of *Trichogramma* spp.**

| Variable | PC1 | PC2 | PC3 | PC4 | PC5 | PC6 | PC7 | PC8 | PC9 | PC10 |
|---|---|---|---|---|---|---|---|---|---|---|
| BL | 0.201 | -0.110 | -0.118 | 0.135 | 0.243 | -0.075 | -0.006 | 0.167 | 0.259 | -0.234 |
| HL | 0.249 | -0.002 | 0.269 | 0.005 | -0.088 | 0.145 | -0.034 | 0.191 | -0.053 | -0.061 |
| HW | 0.254 | -0.017 | 0.244 | 0.027 | -0.103 | 0.130 | 0.020 | 0.241 | -0.099 | -0.102 |
| EW | 0.296 | 0.078 | 0.097 | -0.036 | 0.000 | -0.024 | -0.053 | 0.109 | -0.018 | -0.072 |
| MS | 0.156 | -0.103 | 0.068 | 0.183 | -0.179 | -0.067 | 0.266 | 0.452 | 0.132 | -0.043 |
| FL | 0.162 | 0.245 | 0.162 | 0.202 | 0.172 | 0.034 | 0.091 | -0.115 | 0.009 | 0.028 |
| FW | 0.179 | -0.206 | 0.066 | -0.045 | 0.060 | -0.356 | 0.165 | -0.287 | -0.205 | -0.003 |
| FHL | -0.086 | 0.235 | -0.301 | 0.119 | -0.081 | 0.010 | 0.118 | 0.004 | -0.005 | -0.010 |
| FWL | 0.221 | 0.121 | -0.178 | 0.246 | 0.058 | -0.097 | 0.180 | -0.071 | 0.072 | -0.019 |
| FWW | 0.230 | -0.037 | -0.167 | 0.317 | 0.098 | -0.045 | 0.108 | -0.104 | 0.171 | -0.057 |
| MFW | -0.163 | -0.182 | 0.126 | -0.111 | 0.065 | 0.260 | 0.322 | -0.041 | 0.337 | 0.145 |
| HWL | 0.245 | -0.024 | 0.034 | -0.191 | -0.075 | -0.139 | 0.065 | 0.347 | 0.008 | 0.081 |
| HWW | 0.141 | -0.186 | -0.059 | 0.215 | -0.163 | 0.337 | 0.188 | -0.014 | -0.284 | 0.264 |
| MHW | 0.081 | 0.238 | 0.087 | -0.246 | -0.040 | 0.040 | 0.250 | -0.035 | 0.124 | -0.215 |
| HTL | 0.156 | 0.071 | 0.152 | 0.221 | -0.330 | -0.222 | -0.142 | -0.144 | 0.391 | 0.241 |
| HTW | 0.159 | -0.091 | -0.201 | -0.116 | 0.061 | 0.200 | 0.413 | -0.106 | -0.024 | 0.003 |
| GCL | 0.292 | 0.051 | 0.004 | -0.077 | 0.107 | 0.088 | -0.257 | -0.154 | 0.142 | 0.161 |
| GCW | 0.210 | 0.019 | -0.323 | -0.248 | -0.100 | -0.006 | -0.025 | 0.070 | 0.025 | 0.024 |
| CTG | -0.031 | 0.331 | -0.074 | 0.022 | -0.078 | -0.108 | 0.008 | -0.004 | -0.214 | 0.139 |
| AL | 0.248 | 0.056 | -0.083 | -0.167 | -0.004 | 0.210 | -0.254 | -0.142 | 0.188 | 0.309 |
| RS1 | 0.091 | -0.209 | -0.231 | -0.201 | -0.162 | 0.056 | -0.066 | -0.011 | 0.067 | -0.114 |
| RS2 | 0.037 | -0.058 | 0.016 | 0.095 | -0.437 | 0.308 | -0.163 | -0.297 | -0.091 | -0.605 |
| RM | 0.106 | 0.176 | 0.043 | -0.075 | -0.164 | 0.050 | 0.333 | -0.359 | 0.223 | -0.020 |
| RR | 0.053 | 0.286 | 0.116 | -0.034 | -0.311 | -0.043 | 0.071 | -0.101 | -0.129 | 0.222 |
| NFH | 0.150 | -0.005 | 0.377 | -0.201 | -0.028 | 0.036 | -0.033 | -0.017 | -0.115 | -0.042 |
| FHL/FW | -0.151 | 0.257 | -0.192 | 0.099 | -0.072 | 0.242 | -0.065 | 0.193 | 0.127 | -0.020 |
| FL/FW | -0.067 | 0.300 | 0.013 | 0.129 | 0.031 | 0.333 | -0.121 | 0.210 | 0.157 | 0.007 |
| FL/HTL | 0.090 | 0.241 | 0.091 | 0.095 | 0.399 | 0.175 | 0.188 | -0.042 | -0.236 | -0.118 |
| FWL/FWW | -0.044 | 0.292 | 0.011 | -0.180 | -0.089 | -0.084 | 0.111 | 0.089 | -0.214 | 0.074 |
| FWW/MFW | 0.218 | 0.124 | -0.185 | 0.229 | 0.017 | -0.201 | -0.198 | -0.025 | -0.189 | -0.152 |
| HWW/MHW | 0.055 | -0.258 | -0.085 | 0.285 | -0.113 | 0.221 | -0.014 | -0.000 | -0.278 | 0.318 |
| GCL/GCW | -0.110 | -0.037 | 0.382 | 0.261 | 0.176 | 0.052 | -0.089 | -0.141 | 0.086 | 0.017 |
| GCL/HTL | 0.232 | 0.017 | -0.088 | -0.221 | 0.317 | 0.234 | -0.206 | -0.080 | -0.078 | 0.036 |

(Ratio of hindwings width and maximum marginal length of the hind wing) are negatively correlated variables with PC2. As per the Table 12, both PC1 and PC2 together represent more than 48% of the observations so these characters (EW, GCL, CTG, FW, RS1, FL/FW and HWW/MHW) can be significantly used for the morphometric based differentiation in males of *Trichogramma* spp. which has also adopted in most of the identification work carried out by earlier researchers [34, 42, 52]. Thiruvengadam *et al.* [53] have worked on the molecular taxonomy of *Trichogramma* spp. available from India. They reported that, *T. danaidiphaga* is most distant species among other *Trichogramma* spp. In the present study, similar observation was recorded for *T. danaidiphaga* (Figs 13 & 15).

Similar to this, Table 14 represents the PCA eigenvalues in females of *Trichogramma* spp. for individual characters. In PC1, the characters like OL (Ovipositor length), OL/ACL (ratio of ovipositor length and Antennal club length), FWW/MHW (ratio of forewing width and

**Table 14. Eigenvectors of PCA for the females of *Trichogramma* spp.**

| Variable | PC1 | PC2 | PC3 | PC4 | PC5 | PC6 | PC7 | PC8 | PC9 | PC10 |
|---|---|---|---|---|---|---|---|---|---|---|
| BL | 0.183 | -0.066 | -0.295 | -0.058 | -0.138 | -0.076 | -0.223 | -0.267 | -0.141 | -0.227 |
| HL | 0.246 | -0.150 | -0.108 | -0.124 | 0.017 | 0.220 | 0.060 | -0.040 | 0.362 | 0.466 |
| HW | 0.243 | -0.191 | -0.149 | -0.137 | 0.160 | 0.089 | -0.180 | -0.019 | 0.234 | 0.276 |
| EW | 0.279 | -0.175 | -0.055 | 0.138 | 0.002 | 0.162 | -0.097 | 0.018 | 0.133 | -0.205 |
| MS | 0.035 | 0.001 | -0.266 | -0.011 | 0.010 | 0.408 | 0.359 | 0.110 | -0.080 | -0.495 |
| ACL | -0.119 | -0.298 | -0.241 | -0.170 | 0.311 | 0.160 | -0.056 | -0.064 | -0.151 | 0.015 |
| ACW | 0.130 | 0.221 | 0.044 | -0.267 | 0.031 | 0.328 | -0.359 | 0.354 | 0.025 | -0.111 |
| FWL | 0.033 | -0.218 | 0.361 | -0.188 | 0.174 | -0.118 | 0.123 | 0.062 | -0.028 | -0.204 |
| FWW | 0.168 | 0.097 | 0.271 | -0.222 | 0.174 | 0.116 | 0.395 | -0.184 | -0.151 | -0.049 |
| MFW | -0.255 | 0.183 | 0.131 | -0.139 | 0.084 | 0.251 | 0.121 | -0.209 | 0.136 | 0.056 |
| HWL | 0.157 | -0.249 | 0.122 | -0.071 | -0.241 | 0.064 | 0.392 | 0.189 | 0.208 | 0.069 |
| HWW | 0.063 | 0.119 | -0.119 | -0.274 | 0.319 | -0.344 | 0.124 | 0.110 | 0.473 | -0.241 |
| MHW | -0.157 | -0.205 | 0.292 | -0.107 | 0.032 | 0.236 | -0.150 | -0.086 | 0.242 | -0.196 |
| HTL | -0.038 | -0.133 | -0.084 | -0.533 | -0.179 | 0.008 | -0.018 | -0.199 | -0.136 | 0.069 |
| HTW | 0.118 | -0.014 | 0.166 | -0.369 | 0.191 | -0.148 | -0.181 | 0.044 | -0.408 | 0.113 |
| OL | 0.352 | -0.107 | -0.029 | -0.007 | -0.010 | 0.073 | -0.061 | -0.227 | -0.090 | -0.027 |
| RS1 | 0.073 | 0.140 | 0.256 | 0.260 | 0.377 | 0.037 | 0.045 | 0.032 | -0.082 | 0.273 |
| RS2 | 0.111 | 0.142 | 0.236 | 0.051 | 0.270 | -0.037 | -0.194 | -0.416 | 0.178 | -0.245 |
| RM | 0.142 | -0.223 | 0.209 | 0.049 | 0.046 | 0.025 | 0.080 | 0.396 | -0.188 | 0.033 |
| RR | -0.051 | -0.334 | 0.211 | 0.035 | -0.166 | -0.141 | 0.007 | -0.149 | 0.151 | -0.090 |
| ACL/ACW | -0.170 | -0.337 | -0.142 | 0.112 | 0.140 | -0.163 | 0.233 | -0.290 | -0.091 | 0.078 |
| OL/ACL | 0.332 | 0.089 | 0.134 | 0.115 | -0.189 | -0.052 | -0.017 | -0.133 | 0.045 | -0.040 |
| HTL/ACL | 0.095 | 0.207 | 0.197 | -0.210 | -0.471 | -0.176 | 0.036 | -0.075 | 0.065 | 0.027 |
| FWL/FWW | -0.144 | -0.332 | 0.087 | 0.049 | 0.012 | -0.244 | -0.291 | 0.254 | 0.124 | -0.158 |
| FWW/MFW | 0.331 | -0.112 | -0.001 | 0.006 | 0.011 | -0.169 | 0.082 | 0.070 | -0.223 | -0.088 |
| HWW/MHW | 0.152 | 0.204 | -0.282 | -0.100 | 0.173 | -0.375 | 0.187 | 0.128 | 0.100 | -0.007 |
| OL/HTL | 0.327 | -0.025 | 0.029 | 0.266 | 0.088 | 0.058 | -0.044 | -0.093 | -0.010 | -0.056 |

maximum length of marginal hind wings) and OL/HTL (ratio of ovipositor length and hind tibial length) shows positive but weak association with PC1 (≥0.300) which means that change in these variables will lead to the change in PC1; These association either positive or negative are weak as there are large no. of variables and components. There are only few characters which can solely play major role in identification of any species, but together few characters can. In addition, female characters ACW (antennal club width), HTL/ACL (ratio of hind tibial length and antennal club length), HWW/MHW (ratio of hind wings width and maximum marginal length of hind wings) shows positive (weak) correlation (≥0.200), whereas, RR (setae between RS2 and RM), ACL/ACW (Ratio of Antennal club length and width) and FWL/FWW (ratio of forewing length and width) represent negative (weak) correlation (≥-0.300) with PC2. As per the result in Table 12, the first two PCs together represent the variability of 40% in observations of females. So, these characters are related with both PC1 and PC2 either by positive or negative correlation. Thus, they could be significantly used to identify and differentiating of female in *Trichogramma* spp. based on morphometric data base.

Fig 16 represents the correlation circle of characters in both male and females of *Trichogramma* spp. The component 1 and 2 (F1 & F2) can interpret around 48% of initial information of the dataset in males, whereas, almost 40% in females (Fig 16). The vectors in Fig 16 are characters including ratio characters considered for the morphometric analysis of male and

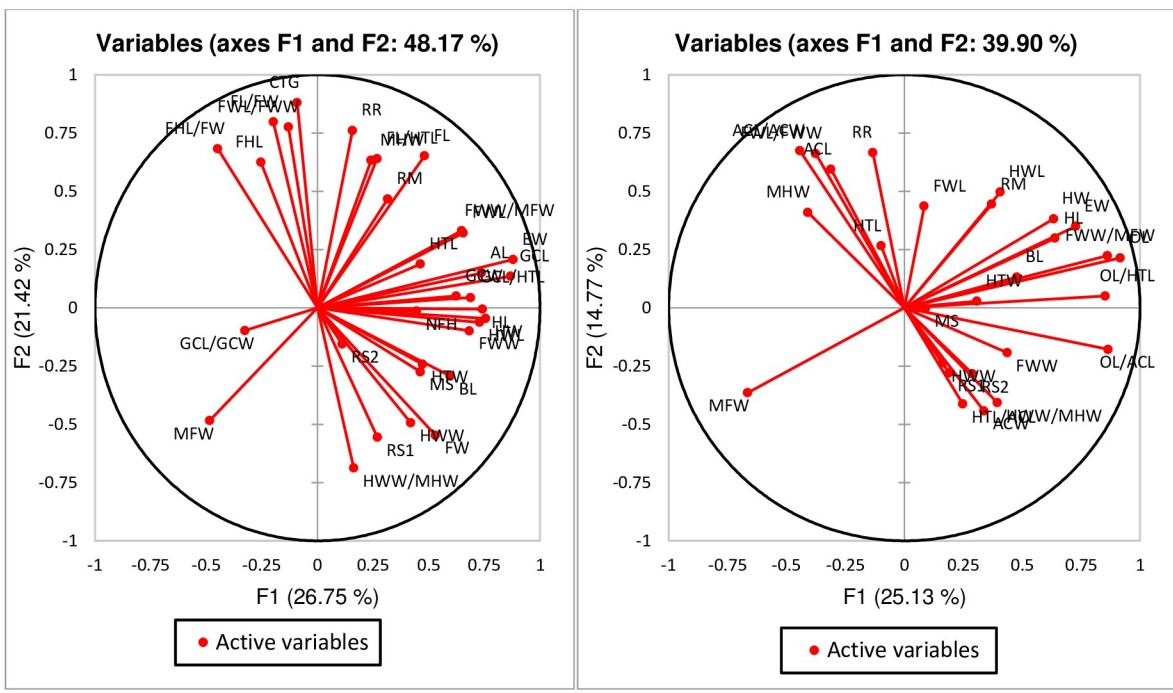

**Fig 16. Correlation circle of characters in male and female *Trichogramma* spp. on different principal components.**

female *Trichogramma* spp. The length of the vector is correlated with the significance of the characters. In males, most of the characters are having long vector line which means majority of the characters are able to represent by F1 and F2 axis. Some of the vectors (characters) such as GCL/GCW, RS2, HTL and NFH were shown with shorter vector line which means they are best described in other PCA dimensions. Similarly, most of the vectors in female *Trichogramma* spp. were represented significantly by both F1 and F2 PCA axis but few characters like HTL, FWW, HTW, BL, MS, HWW, RS1 and RS2 can be best described in other PCA dimensions (Fig 16). The degree between the vectors line represent the relation between the characters. Vectors with obtuse angles are negatively correlated and with acute angles are positively correlated. Also, vector separated by right angles shows correlation of neither positive nor negative. So, it can be interpreted that characters like RR, RM, HTL, AL, GCL, HTL and others show a good significant positive correlation among each other. Similarly, characters like FWL, HWL, RM, HW, HL, BL, FWW/MFW and OL/HTL shows a varying degree of significant positive correlation among each other.

Fig 17 represents the species wise segregation on both first and second principal component dimensions. In the graph, individuals of *Trichogramma* spp. shows a similar observation pattern within the species and thus clustered at specific site in of the graph. If a monoculture species is clustered together then it shows the uniqueness of the characters and ability to be significantly different from other species. For males, it can be seen that few observations in *T. brassicae* are significantly unique from other species of *Trichogramma* spp. as they are clustered on left top corner of the graph (Fig 17). It also depicts that species, *T. chilotraeae* and *T. danausicida* shows few similar characters due to which they are clustered together. Similar observation was also drawn from the female *Trichogramma* spp. graph on Fig 17. In females, *T. brassicae* and *T. chilotraeae* shows similar pattern for few characters. Fig 18 shows the biplot of the variables and observations simultaneously. Both males and females of *Trichogramma*

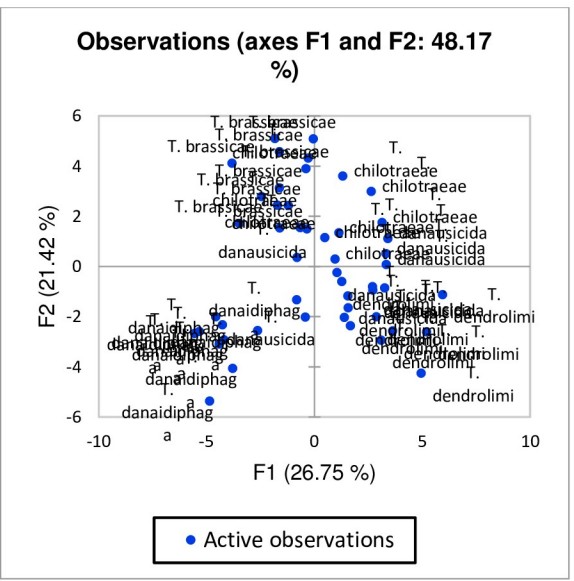
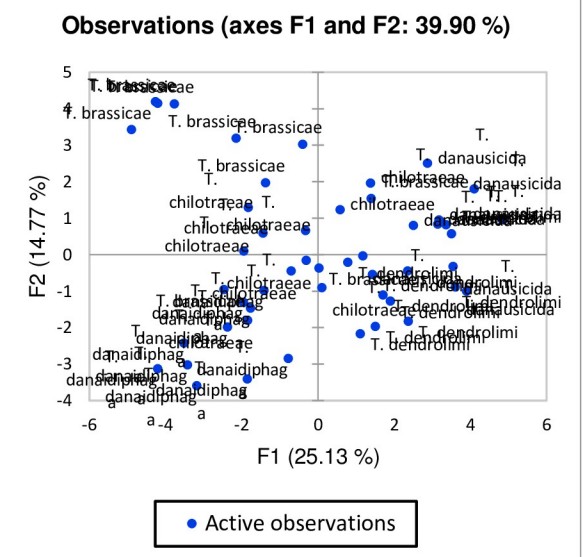

**Fig 17. Plots of individuals in male and female *Trichogramma* spp. on different principal components.**

spp. are indicated on the maps. Here it can be identified that males of *Trichogramma* spp. like *T. danausicida* and *T. chilotraeae* favour significant correlation in observation characters like RR, RM, EW, FWW/MFW, GCL, GCL/HTL and MFW. Character MFW (Maximum marginal length on forewings) is peculiar for *T. danaidiphaga* which can be useful for separating this species from others. Similar to this, characters like HWW/MHW, RS1, HWW, FW, HL, NFH, FWW, HTW, MS and BL are important for separating and identifying *T. dendrolimi* from *T. brassicae*, *T. danaidiphaga*, *T. danausicida* and *T. chilotraeae*. Though, GCL/GCW is able to segregate *T. danaidiphaga* from other *Trichogramma* spp. but it can be better justified in other PCs. Also, characters like CTG, FHL, Fore wigs characters such as FWL/FWW, FHL and FHL/FW are best correlated for *T. brassicae* and *T. chilotraeae*.

In females, characters FWL, RM, HWL, OL, OL/HTL, HL, BL, HW, HTW and FWW/MFW are correlated best in both *T. danausicida* and *T. chilotraeae* (Fig 18). Individuals of *T. danaidiphaga* can be best identify by using character MFW. Characters like OL/ACL, HWW/HWL, RS1, RS2, HWW, FWW, ACW and HTL are able to separate the individuals of *T. dendrolimi* from other studies *Trichogramma* spp. Also, Antennal characters (ACL, ACL/ACW), FWL/FWW, MHW and RR are significantly correlated in both *T. brassicae* and *T. chilotraeae*. So, it can be concluded that there are few characters of males and females which can be significantly used for taxonomic identification and differentiation of species in *Trichogramma*. Thus, morphometric tool can be helpful for authentic identification of different species and able to differentiate two different species in *Trichogramma* genus.

## Conclusion

The present study was carried out for identification and differentiation of five species in *Trichogramma* genus by using morphometrics database and technique. PCA was employed to analysis the morphometric observations. It was observed that, most of the characters of male such as BL, HW, EW, FHL, MFW HWL, HTW, CTG, AL, setae in RS2, RM and RR were able to successfully identify and separate the one species from other (Figs 19 and 16–18). Apart from

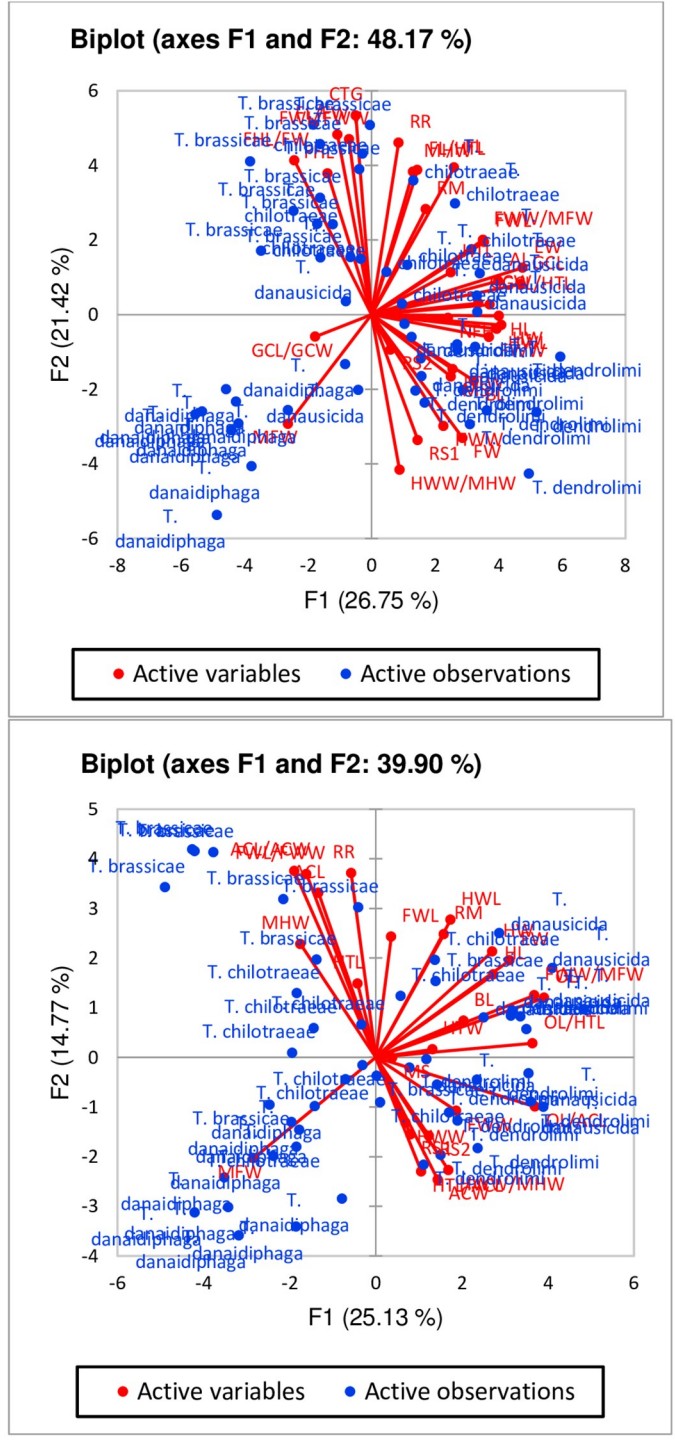

**Fig 18. Biplot of different species of *Trichogramma* with active characters on different principal components.**

these traditional characters like the ratios of FHL/FW, FWL/FWW, HWW/MHW and GCL/
GCW (Figs 20 and 16–18) were also made significant contribution in species identification
and differentiation. Similar to this, in females, BL, HL, EW, ACL, ACW, MFW, HWL,
MHW, HTW, OL, setae in RS1 and RM (Figs 21 and 16–18) were identified as important

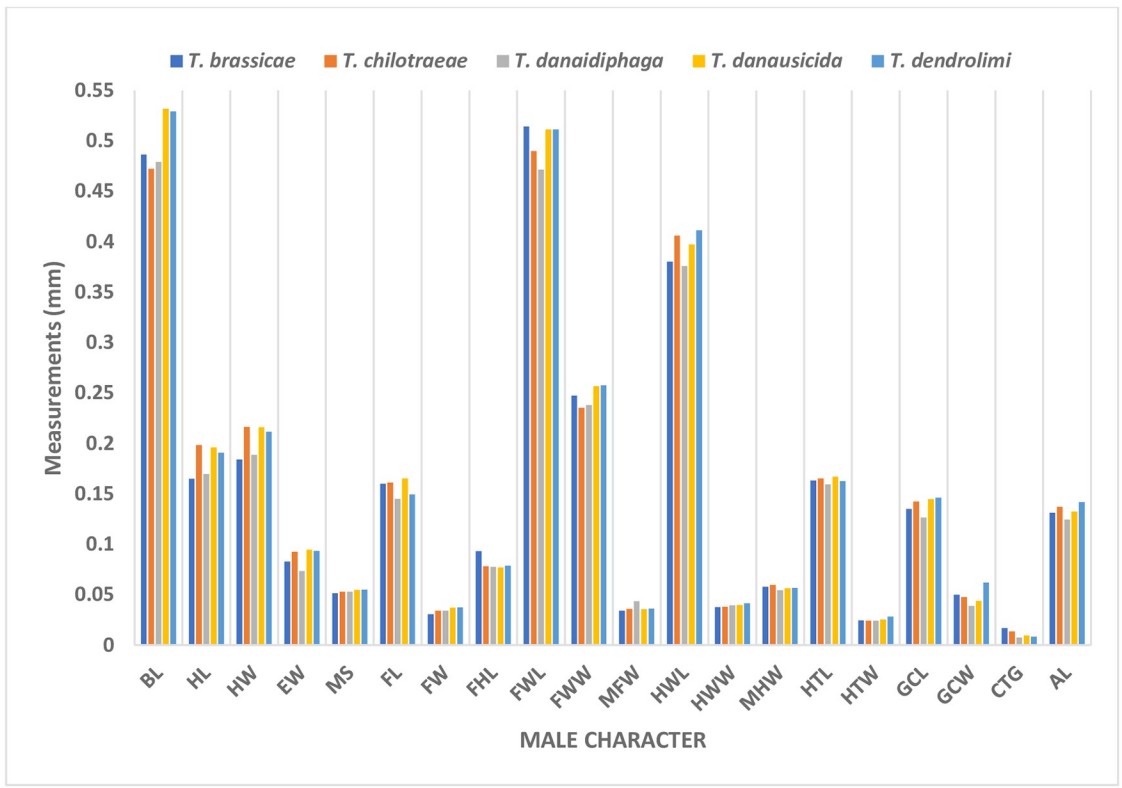

**Fig 19. Comparative morphometrics of male *Trichogramma* spp.**

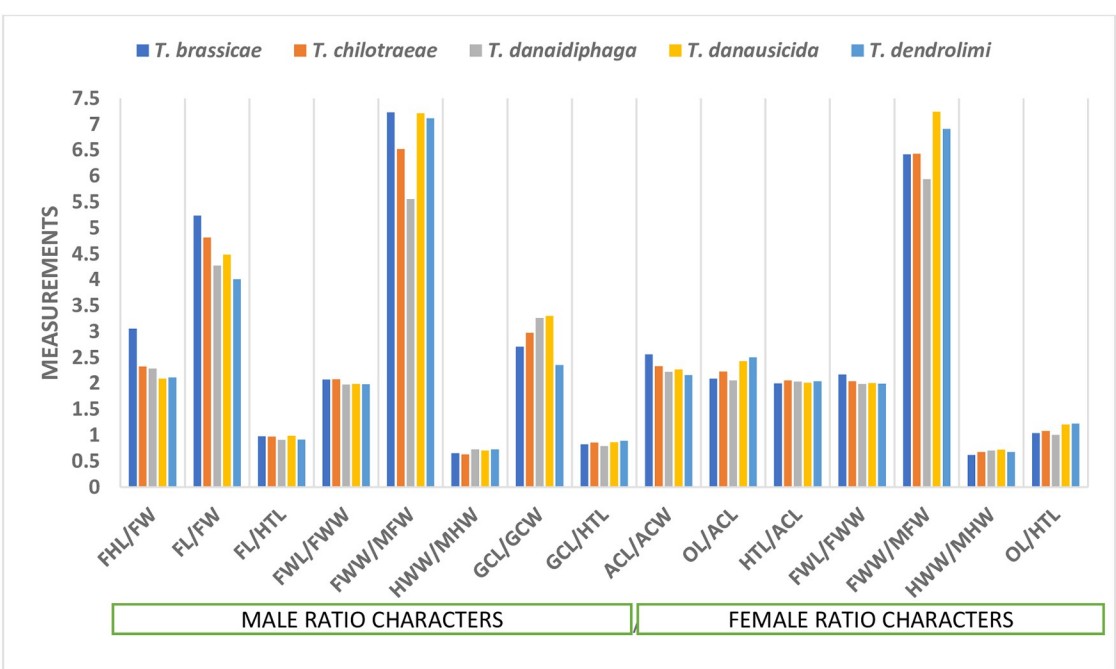

**Fig 20. Comparative morphometrics of ratio characters in *Trichogramma* spp.**

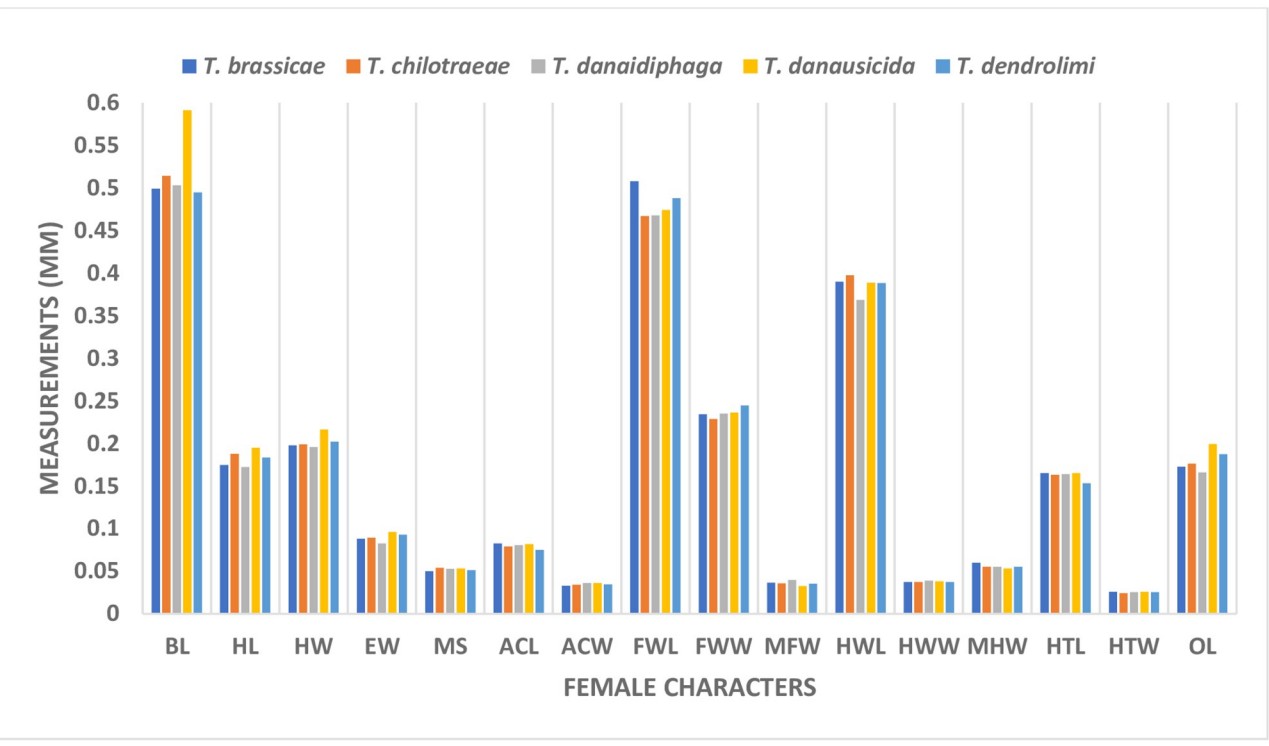

**Fig 21. Comparative morphometrics of female *Trichogramma* spp.**

morphometric characters which has potential for differentiation among species. Also, few ratio characters such as OL/ACL, FWW/MFW, FWL/FWW and OL/HTL (Figs 20 and 16–18) were able to be used in morphometric based differentiation of the species. The research findings of the present work include successful identification of the morphometric characters in both male and female *Trichogramma* spp. for authentic taxonomic identification. This study also confirms that few characters of female which may be useful in taxonomy can identified using morphometric technique with more accuracy. The present study will explore the differentiation of other *Trichogramma* species reported across the boundaries. The study will act as a reference-document consisting of protocol, guidelines, species-based information, morphometric data, key and others for the correct identification of the species for better use in biological control programme.

## Acknowledgments

We would like to express our gratitude to Director, Forest Research Institute, Dehradun, Uttarakhand (India) for providing the necessary research facilities. We are also like to thank Regional Director, Forest Survey of India, Central Zone, Nagpur for encouragement and allow me to write the manuscript and to provide the necessary facility. We are also greatly indebted to the authorities of NBAIR (Bangalore, India) for providing the culture of *Trichogramma* species for the study. We are also very grateful to English editor (Dr. Ashok Kumar Dhakad) Scientist, Punjab Agricultural University Ludhiana, India whose constructive comments and editing have greatly improved this paper both grammatically and typologically as well.

## Author Contributions

**Conceptualization:** Salman Khan, Mohd. Yousuf.

**Formal analysis:** Salman Khan, Mohsin Ikram.

**Investigation:** Salman Khan, Mohsin Ikram.

**Methodology:** Salman Khan, Mohd. Yousuf.

**Project administration:** Mohd. Yousuf.

**Software:** Salman Khan, Mohsin Ikram.

**Validation:** Salman Khan, Mohsin Ikram.

**Writing – original draft:** Salman Khan.

**Writing – review & editing:** Salman Khan.

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
