## [Decision Letter · Decision Letter 0]

18 Mar 2020

PONE-D-20-03271

Morphometric based differentiation in five different species of genus Trichogramma

PLOS ONE

Dear Dr. KHAN,

Thank you for submitting your manuscript to PLOS ONE. After careful consideration, we feel that it has merit but does not fully meet PLOS ONE’s publication criteria as it currently stands. Therefore, we invite you to submit a revised version of the manuscript that comprehensively addresses the points raised during the review process by the two reviewers, both of which have made a number of highly pertinent comments that need to be addressed in full in the revised manuscript.

We would appreciate receiving your revised manuscript by May 02 2020 11:59PM. To enhance the reproducibility of your results, we recommend that if applicable you deposit your laboratory protocols in protocols.io, where a protocol can be assigned its own identifier (DOI) such that it can be cited independently in the future. For instructions see: http://journals.plos.org/plosone/s/submission-guidelines#loc-laboratory-protocols

We look forward to receiving your revised manuscript.

Kind regards,

Michael Schubert

Academic Editor

PLOS ONE

Reviewers' comments:

Reviewer's Responses to Questions

**Comments to the Author**

1. Is the manuscript technically sound, and do the data support the conclusions?

Reviewer #1: Yes

Reviewer #2: No

2. Has the statistical analysis been performed appropriately and rigorously? 

Reviewer #1: Yes

Reviewer #2: Yes

3. Have the authors made all data underlying the findings in their manuscript fully available?

Reviewer #1: Yes

Reviewer #2: Yes

4. Is the manuscript presented in an intelligible fashion and written in standard English?

Reviewer #1: No

Reviewer #2: Yes

5. Review Comments to the Author

Reviewer #1: I have carefully read the manuscript titled “Morphometric based differentiation in five different species of genus Trichogramma”. In years, success of biological control is infected dramatically by specific species identification especially for Trichogramma biocontrol. Hence, insightful identification on Trichogramma wasp is valuable to determine suitable approaches for field release. The sheer amount of work that went into doing the identification and analyses for a rarely described metrics, and on this count alone it deserves publication. However, I think there are some flaws and rough indications for this paper to be published in this journal. I recommend resubmission after major revision.

The current manuscript uses morphometric approaches to examine the importance traits of several egg parasitoids. While the idea and results are interesting and important, much work remains to be done in preparing the narrative to meet proper standards of English syntax and grammar. I believe this manuscript will require careful proofing by an English first language writer who is familiar enough with the scientific content to properly present the technical parts of the narrative.

Additionally, the abstract did not indicate the importance of their work, methods description and presentation of results, discussion are a little bit confusing and need more clarify conclusion and logical linkage. Meanwhile, I think that they need to really lay out the limitations of their study and to include the relative molecular study more precisely (for example ms published by Muhammad et al. 2013 in Biocontrol doi: 10.1007/s10526-013-9509-z; Venkatesan et al. 2016 in Biological Control doi: 10.1016/j.biocontrol.2016.07.005) in discussion and conclusion paragraph as well. This will add value to the paper. Some of important reviews/ref. are also a bit outdated. Please see attached comments for specifics.

Specific comments:

1. Please clarify the following sentence “In morphometrics, the stressed are mostly on the genitalia characteristics, antennae, forewings of the males for the identification of different species of Trichogramma”

2. Introduction, the reasons for current article and selection of five Trichogramma species are ambiguous and may need more indication.

3. More details are needed in the statements “The specimens of Trichogramma were stored in 70% Ethanol followed by soaking into Potassium Hydroxide (KOH) 10%.” For how long? And is there any temperature treatment?

4. “After treatment with Ethanol, the specimens were kept in Clove oil for a few minutes” Please list the method more precisely.

5. Page 11, figure 0, Please add on the figure legends for the numbers.

6. Page 15, Genital capsule should be presented in higher resolution photo, and more morphological traits can be described as well.

I am not familiar with the procedures used in the Morphometric measurements. However, their result and conclusion need to be more clear and well-organized to strengthen the importance and the efforts of current work.

I hope these comments are useful.

Reviewer #2: The structure of the manuscript should be reorganized. The manuscript is a bit confused about the goals and results. So, what is the reason for the study? This must be clear. I also suggest more carefully review the manuscript. I recommend that authors to analyze recommendations below:

1. Introduction - It´s no clear about the aims of this study. What the real propose of the morphometric differentiation of species?

2. What is the reason for including the figures on the morphology? The morfological characterization was one of the goals of the manuscript?

3. The morphological description was not included in the methodology.

4. What is the procedure used to identify species?

5. It is not clear in the methodology which number of specimens used for morphometry.

6. What is the reason for the inclusion of females, if they can’t be used for morphological differentiation?

7. The figure 0 needs to be improved, images are incipient and the lines aren´t precise. I don't really see a need for it, since the characters have been described.

8. Cite a source for terminology.

9. The results are confused in relation to the individual descriptions and the tables with the descriptions. It is not necessary to repeat data.

10. I suggest evaluating the results presented. Is it necessary to present all the morphometric descriptions? Or present the joint results for the species in graphs or tables, then interpret and discuss the results.

11. It is not necessary to analyze the characters of the females, because it is not possible to separate them morphologically. How were they identified?

12. In the results, it does not indicate which characters were discriminated to separate the females.

13. The images need to be improved (Examples: 1a-A, 1a-B, 1a-G, 2b-A, 3a-A, 3a-B, 3a-G, 3b-A, 3b-B, 3b-G, 4a-A, 4a-G,

14.What is the conclusion?

6. PLOS authors have the option to publish the peer review history of their article (what does this mean?). If published, this will include your full peer review and any attached files.

Reviewer #1: No

Reviewer #2: No

---

## [Author Response · Author response to Decision Letter 0]

30 Mar 2020

Comments to Reviewer 1: 

1. The sentence “In morphometrics, the stressed are mostly on the genitalia characteristics, antennae, forewings of the males for the identification of different species of Trichogramma” have been re-written for better clarification.

2. In Introduction, the importance of current work has been added with description.

3. The reason for selecting these five species of Trichogramma for morphometric analysis have been added in the material and method section.

4. The material and method section have been reformed with addition on few information such as duration of any treatments and temperature treatment. 

5. Figure legend for the numbers are already mentioned in the “Morphometric observation” part of Material and Methods section. 

6. Figure 1a on page 15 is being attached in separate file as fig 2a for better resolutions. Also, morphometric traits have been incorporated and described in details for each species in their diagnosis section. All morphometrics figures are attached as .tiff file format for better resolution.

7. The Abstract section has been revised by providing some information about the reason for current study, methodology and results.

8. The present study is limited to the morphometric studies based on morphological characters only. Molecular analysis was not carried out in the present study. Molecular is advanced tool but traditional methodology is also a reputed tool for separating different species. In future, I will work on molecular analysis of insects but for now my study is restricted to morphological aspects only. 

9. The results and discussion part of the manuscript has been completely revised as per the reviewer comments. The flaws in English language of the manuscript were removed at authors’ best level. The haphazard of the results and discussion part are revised to make a meaningful and understandable results.

10. The conclusion is added in the manuscript as per the reviewer comments.

 

Comments to Reviewer 2: 

1. The objective of the study is added in the Introduction part of the manuscript. Also, the importance of the study is discussed in details. 

2. The morphological characters are the basis of present study. The Figure 0 is the demonstration of considered part/character of any body part of either male or female. Also, few characters have been considered for the first time in morphometric analysis. The ratios of few characters were calculated to identify their applicability in morphometric separation of Trichogramma spp. So, to avoid confusion among the readers for any abbreviation or characters full name, the Figure 0 is given in the material and methods section. Also, figure 0 will be attached in separate .tiff file as the image is good quality but in MS Word, it become little blur. 

3. The morphological diagnosis for all the five species of Trichogramma are incorporated in Results and Discussion section. 

4. The cultures obtained from the laboratory were identified using the literature available. The details are provided in the Material and Method section.

5. The no. of specimens used for morphometry has been included in the manuscript.

6. The females of Trichogramma spp. cultures were identified using the literature available. Hind tibial length and length of aedeagus and apodemes were compared with the ovipositor to identify and confirm the species.

7. As per the earlier researchers, females’ characters are less useful in identification and separation of Trichogramma spp. In present study, it was attempted to check whether the female characters could be used for morphological differentiation using morphometry. In this study, few characters of females were found significant in separating studied species. These characters will be highlighted in the manuscript and separate paragraph will be added to show them significantly. 

8. References for the terminology used are properly cited in the manuscript. E.g. Yousuf and Shafee, 1988; Hassan and Yousuf, 2007 and Khan et al., 2019.

9. All the images of morphometric characters of Trichogramma spp. will be uploaded as .tiff file to avoid blurriness. 

10. The morphometric observations of the Trichogramma spp. were repetitive in table also. So, from observation, repetitive paragraphs are removed. Only characters, which are discussed with previous research work have been shown. 

11. The conclusion is added in the manuscript as per the reviewer comments. 

12. The results and discussion part of the manuscript has been completely revised as per the reviewer comments. The flaws in English language of the manuscript were removed at authors’ best level. The haphazard of the results and discussion part are revised to make a meaningful and understandable results.

13. In results and discussion part, earlier morphometric observation was given for individual species. After revision, apart from little individual observation on species, a comparative study among the species is also included using the graphs.

---

## [Decision Letter · Decision Letter 1]

20 May 2020

PONE-D-20-03271R1

Morphometric based differentiation in five different species of genus Trichogramma

PLOS ONE

Dear Dr. KHAN,

Thank you for submitting your manuscript to PLOS ONE. After careful consideration, we feel that it has merit but does not fully meet PLOS ONE’s publication criteria as it currently stands. Therefore, we invite you to submit a revised version of the manuscript that addresses the points raised during the review process.

We would appreciate receiving your revised manuscript by Jul 04 2020 11:59PM. To enhance the reproducibility of your results, we recommend that if applicable you deposit your laboratory protocols in protocols.io, where a protocol can be assigned its own identifier (DOI) such that it can be cited independently in the future. For instructions see: http://journals.plos.org/plosone/s/submission-guidelines#loc-laboratory-protocols

We look forward to receiving your revised manuscript.

Kind regards,

Michael Schubert

Academic Editor

PLOS ONE

Reviewers' comments:

Reviewer's Responses to Questions

**Comments to the Author**

1. If the authors have adequately addressed your comments raised in a previous round of review and you feel that this manuscript is now acceptable for publication, you may indicate that here to bypass the “Comments to the Author” section, enter your conflict of interest statement in the “Confidential to Editor” section, and submit your "Accept" recommendation.

Reviewer #1: All comments have been addressed

2. Is the manuscript technically sound, and do the data support the conclusions?

Reviewer #1: Yes

3. Has the statistical analysis been performed appropriately and rigorously? 

Reviewer #1: Yes

4. Have the authors made all data underlying the findings in their manuscript fully available?

Reviewer #1: Yes

5. Is the manuscript presented in an intelligible fashion and written in standard English?

Reviewer #1: No

6. Review Comments to the Author

Reviewer #1: I have carefully read the manuscript titled “Morphometric based differentiation in five different species of genus Trichogramma”. Most of comments were carefully revised.

I believe this manuscript will be more sound by an English first language writer who is familiar enough with the scientific content to properly present the technical parts of the narrative.

I think that they can compare their morphometric results with molecular identification, which can be done through NCBI database (for example ms published by Muhammad et al. 2013 in Biocontrol doi: 10.1007/s10526-013-9509-z; Venkatesan et al. 2016 in Biological Control doi: 10.1016/j.biocontrol.2016.07.005) This will add value to the paper.

Specific comments:

Introduction

"In the natural

system, the insect population is being regulated by various factors like temperature, sunlight, crop composition and others." This sentence has weak connection with its context.

Results

"shows positive correlation (≥0.200)" correlation is positive but not very sigificant, more statistics or indication is needed.

Conclusion

"So, that a document can be prepared to identify the correct species

for better use in biological control programme." More details are needed in this sentence, like what kind of document, or any standard protocol to follow?

7. PLOS authors have the option to publish the peer review history of their article (what does this mean?). If published, this will include your full peer review and any attached files.

Reviewer #1: No

---

## [Author Response · Author response to Decision Letter 1]

3 Jul 2020

1. The manuscript is edited with the English first language writer who have greatly improved this paper both grammatically and typologically. 

2. All the comments suggested by the reviewer are accepted by the author(s). We are thankful to the reviewer for his exhaustive comments on the manuscript which was very essential to make this article a good read.

3. author(s) have also tried to remove all flaws in the manuscript and modified the manuscript as per the reviewer guidelines. 

4. As per the reviewer comments in Introduction part, sentence “In the natural system, the insect population is being regulated by various factors like temperature, sunlight crop composition and others” has been Omitted and sentence is re-formed. 

5. As reviewer suggested about the elaboration and more indication on “shows positive correlation (≥0.200)”, we have mentioned about the intensity of correlation. Also, detailed description has been added in the manuscript about the statistics behind these PCs (Principal Components) and characters association. These correlations are related with different principal components but not with the species itself. As there is large no. of variables so all the variable could not participate in attaining greater association with PCs. Though, some of these characters all together can generate good correlation and this is the objective of the study to finalize and identify the group of characters which can helpful in identification and separation of the species in Trichogramma. 

6. In Conclusion part, reviewer commented on “So, that a document can be prepared to identify the correct species for better use in biological control programme”. The authors appreciated the query and information has been added to the manuscript about the type of document and its content. The sentence has been revised as So that a reference-document consisting of protocol, guidelines, species-based information, morphometric data, key and others can be prepared to identify the correct species for better use in biological control programme. 

7. The reviewer comments have been taken into consideration. Venkatesan et al. (2016) in Biological Control (doi: 10.1016/j.biocontrol.2016.07.005) is actually, Thiruvengadam et al. (2016). The authors have compared the molecular taxonomy work by Thiruvengadam et al. (2016) with the morphological taxonomy work of present study. 

8. The author agrees with the reviewer suggestion regarding the molecular taxonomy of Trichogramma spp. The present study is limited to the identification based on morphology only. In future studies, we will start working on molecular aspect too. Molecular is the latest technological methodology being utilized widely in insect taxonomy. Still, most of the insect taxonomies around the world are based on morphological characters only. These characters are vital in the separation of the different species of any genus. Specifically, in Trichogramma spp. morphological characters such as genitalia length, width, ovipositor length, forewings characteristics are often utilized for its identification by various researchers such as Nagaraja and Nagarkatti (1969), Garcia-Gonzalez; Rohi and Pintureau, 2003; Hernandez-Gonzalez (2009) and Querino et al. (2017). As the taxonomic inferences described by morphology are more reliable than any other method. Even, most of the researchers who have maximum no. of species under their credit are still depends on morphological based taxonomy. Few researchers such as Ranyse B. Querino, Roberto A. Zucchi, J.D. Pinto, Nagaraja and Nagarkatti have primarily chosen morphological based identification in most of their studies.

So, in future studies, the molecular technique will be employed along with the morphological methods but for now, we are limited with morphological character-based identification due to various reason.

Thanking you

---

## [Editor Report · Decision Letter 2]

8 Jul 2020

Morphometric based differentiation among Trichogramma spp.

PONE-D-20-03271R2

Dear Dr. KHAN,

We’re pleased to inform you that your manuscript has been judged scientifically suitable for publication and will be formally accepted for publication once it meets all outstanding technical requirements.

Kind regards,

Michael Schubert

Academic Editor

PLOS ONE

---

## [Editor Report · Acceptance letter]

13 Jul 2020

PONE-D-20-03271R2 

Morphometric based differentiation among *Trichogramma* spp. 

Dear Dr. KHAN:

I'm pleased to inform you that your manuscript has been deemed suitable for publication in PLOS ONE. Congratulations! Your manuscript is now with our production department. 

Kind regards, 

on behalf of

Dr. Michael Schubert 

Academic Editor

PLOS ONE